# Forward-Chaining Temporal Point Process

**Chao Yang** [1]  **Wendi Ren** [1]  **Shuang Li** [1]

## Abstract

Event sequences from complex systems, such as clinical workflows, are often sparse and incomplete. As a result, downstream models are trained on data that only partially captures the underlying dynamics. Synthetic sequence generation can augment real data by filling in missing structure and improving coverage of rare patterns, but generated trajectories must remain realistic, satisfy domain constraints, and allow control. We propose the Forward-Chaining Temporal Point Process (FC-TPP), a framework for constraint-aware and controllable sequence generation in continuous time. FC-TPP maintains an explicit latent symbolic state encoding high-level predicates, which evolves through a differentiable multi-hop forward-chaining operator. Logical rules update the latent state based on recent events, while a temporal point process decoder generates future event times and types conditioned on this evolving state. By tying the generative dynamics to multi-hop reasoning in latent space, FC-TPP incorporates symbolic structure throughout generation rather than relying directly on raw event histories. Experiments on synthetic data and four semi-synthetic/real-world benchmarks—LogiCity, MIMIC-IV, EPIC-100, and IKEA ASM—show that FC-TPP achieves higher generation quality under limited and incomplete data, with stronger constraint adherence and greater controllability than purely neural and prior neuro-symbolic baselines.

## 1. Introduction

In many safety-critical domains—such as healthcare, finance, and security—the availability of real-world event sequences (e.g., patient trajectories, activity traces, financial transactions) is severely limited (Potluru et al., 2023; Wornow et al., 2024). Data are often scarce, privacy-sensitive, and difficult to share. Even when available, observed sequences are noisy or incomplete, providing only partial evidence of how the underlying process evolves. This lack of high-quality data poses a bottleneck for downstream modeling, as modern learning methods rely on broad and reliable coverage of valid behaviors.

Generative models offer a principled way to mitigate this bottleneck. By imputing missing events and synthesizing new trajectories, they can both improve data quality and expand coverage beyond what is observed. However, in safety- and decision-critical settings, usefulness requires more than surface-level realism. Generated sequences must be consistent with domain rules that govern admissible system evolution (e.g., eligibility and exclusion criteria, ordering and timing constraints), and models should provide some handle for controlling how those rules shape generated trajectories. Enforcing constraints only at the output level is insufficient: effective generation requires maintaining a *logically coherent latent state* throughout the generative process, so that inferred system conditions and sampled events remain mutually consistent even when observations are sparse or incomplete.

To address these challenges, we propose **Forward-Chaining Temporal Point Processes (FC-TPP)**, a generative framework that integrates temporal point processes (TPPs) with an explicit *latent-space reasoning layer*. FC-TPP represents the evolving system using a structured *latent symbolic state* whose dimensions correspond to high-level predicates. Rather than treating reasoning as an auxiliary signal added to a neural decoder, FC-TPP adopts a *reasoning-before-generation* paradigm: after incorporating new event evidence, a differentiable multi-hop forward-chaining operator applies symbolic rules to propagate implications and complete the latent state. A TPP decoder then generates future event times and types conditioned on this *rule-completed* latent state, naturally accommodating irregular inter-event times. By coupling generation to multi-hop reasoning in latent space, FC-TPP aligns sampled trajectories with inferred high-level system conditions instead of relying solely on correlations in raw histories.

[1]School of Data Science, The Chinese University of Hong Kong, Shenzhen. Correspondence to: Shuang Li <lishuang@cuhk.edu.cn>.

*Proceedings of the $43^{rd}$ International Conference on Machine Learning*, Seoul, South Korea. PMLR 306, 2026. Copyright 2026 by the author(s).

## 1.1. Literature Review

Our model sits in the broader landscape of neural generative TPPs but differs in how it defines and updates latent state. Autoregressive neural TPPs encode history into a hidden state (e.g., via RNNs (Du et al., 2016; Mei & Eisner, 2017) or Transformers (Zuo et al., 2020; Yang et al., 2025a)) and parameterize the conditional intensity or the distribution of the next time and mark. These models are strong density estimators, yet the hidden state is a generic statistical summary of the raw past and is not designed to track admissible conditions; logical constraints are typically handled, if at all, by masking, rejection sampling, or penalty losses. Likelihood-free and two-sample approaches (e.g., adversarial training with a Wasserstein distance critic (Xiao et al., 2017), RL-based formulations with MMD-style rewards (Li et al., 2018)) improve distributional match but still target observational fit rather than the mechanisms that separate valid from invalid trajectories. Latent-variable and transformation-based generators—including VAE-style TPPs (Yang & Zha, 2024), flow-based models (Mehrasa et al., 2019a; Yang et al., 2025b), and diffusion-based sequence samplers (Lüdke et al., 2023)—introduce latent structure either as low-dimensional codes or as stochastic trajectories in event space, typically optimized to capture multi-modality or global variation. In most cases, this latent structure is not designed as an interpretable summary of system state, and their evolution is governed by unconstrained neural transitions. FC-TPP also uses a latent state, but it is explicitly predicate-based and evolves via forward chaining, making "*reasoning-before-generation*" a structural property of the generator.

Recent work has begun to combine large language models (LLMs) with TPPs, primarily to enrich event representations with textual semantics or to use LLMs as external reasoning modules (Shi et al., 2023). In these hybrids, the LLM typically acts as an encoder or assistant, while the underlying TPP dynamics remain essentially unchanged; any constraints the LLM implicitly "knows" are not turned into a stable internal state that is updated in a controlled, stepwise fashion. FC-TPP can optionally use LLMs to initialize predicate and rule embeddings, but the evolution of the model is governed by a fixed forward-chaining operator on a latent symbolic state, and it is this rule-guided latent dynamics that drives generation.

Finally, several neuro-symbolic TPPs incorporate logical structure by shaping intensities with rule-based components (Li et al., 2020b) or by inducing temporal logic rules to explain observed events (Li et al., 2021; Yang et al., 2024). These models are valuable for interpretability and rule discovery, but the symbolic layer typically lives directly on the observed sequence: rules modulate intensities or provide explanations for past events, rather than defining a separate latent process. FC-TPP takes a different approach: multi-hop forward chaining is the *latent transition operator*. Rules update a predicate-level latent state that summarizes high-level conditions, and the TPP decoder generates future times and types conditioned on this evolving abstract state rather than directly on raw histories. This separation allows FC-TPP to infer latent information (e.g., unobserved regimes or eligibility flags), propagate it over time, and use it to constrain generation, while keeping the latent representation compact and explicitly reasoned about.

**Contributions.** In summary:

1. We formulate **FC-TPP**, a neuro-symbolic latent-state model for generative TPPs in which a predicate-level latent state evolves via differentiable multi-hop forward chaining and drives continuous-time event generation.

2. We introduce a *reasoning-before-generation* architecture in which a rule-completed latent state is produced before each sampling step, so that symbolic structure influences the entire sampling path rather than appearing only as a post-hoc constraint.

3. We empirically show on synthetic data and four semi-synthetic/real-world benchmarks (LogiCity, MIMIC-IV, EPIC-100, and IKEA ASM) that FC-TPP improves generation quality under limited and incomplete data, adheres more closely to domain constraints, and offers more controllable behavior than purely neural and prior neuro-symbolic baselines.

## 2. Background Knowledge

### 2.1. Temporal Point Processes (TPPs)

A (marked) temporal point process models a sequence of events $\{(t_i, m_i)\}_{i=1}^{N}$, where $t_i \in \mathbb{R}^+$ is the event time and $m_i \in \mathcal{M}$ is the event type (mark). Let

$$\mathbf{x}_t := \{(t_j, m_j) \mid t_j < t\}$$

denote the sequence history up to time $t$. The dynamics of the TPP are characterized by the conditional intensity function

$$\lambda(t, m \mid \mathbf{x}_t) = \lim_{\Delta t \to 0} \frac{\mathbb{P}\big(\text{event of type } m \text{ in } [t, t + \Delta t) \mid \mathbf{x}_t\big)}{\Delta t},$$

with total intensity across mark types

$$\lambda(t \mid \mathbf{x}_t) = \sum_{m \in \mathcal{M}} \lambda(t, m \mid \mathbf{x}_t).$$

Governed by the conditional intensity, a TPP generates events in an autoregressive fashion. Given the last event

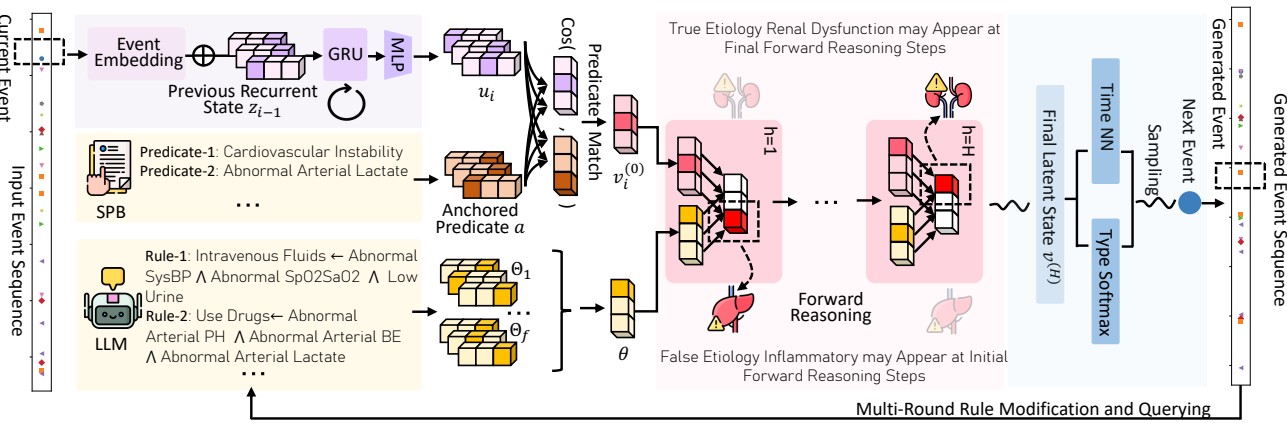

*Figure 1.* Model framework. The background color indicates: "⬛": Neural Pre-Update Module, "⬛": Symbolic Prior Bank and LLM,"⬛": Neuro-Symbolic Reasoning Layer "⬛": Event Decoder.

time $t_i$ and history $\mathbf{x}_{t_i}$, the next event time $T$ has survival function

$$\mathbb{P}(T > \tau \mid \mathbf{x}_{t_i}) = \exp\Big(-\int_{t_i}^{\tau} \lambda(s \mid \mathbf{x}_s)\,ds\Big), \quad (1)$$

$$\mathbb{P}(m = k \mid T = t, \mathbf{x}_t) = \frac{\lambda(t, k \mid \mathbf{x}_t)}{\lambda(t \mid \mathbf{x}_t)}. \quad (2)$$

In many models, the integrated intensity $\int_{t_i}^{\tau} \lambda(s \mid \mathbf{x}_s)\,ds$ does not admit a closed-form inverse, and one typically uses Ogata's thinning algorithm (Ogata, 1981; Rasmussen, 2018) to sample the next event time.

This TPP framework naturally handles *irregularly spaced events and continuous-time dynamics*, making it a flexible decoder for FC-TPP.

## 2.2. Logical Predicates and Forward-Chaining Inference

We view each event history as an element of a sequence space $\mathcal{X}$. For a TPP, we write $\mathbf{x}_t \in \mathcal{X}$ for the history up to time $t$; in this subsection we use $\mathbf{x} \in \mathcal{X}$ generically.

A **predicate** is a Boolean function

$$P : \mathcal{X} \to \{0, 1\},$$

which abstracts a raw event history $\mathbf{x}$ into a high-level symbolic property (e.g., "patient is on anticoagulation"). Let $\mathcal{P}$ denote a finite collection of such predicates.

Domain knowledge is encoded as a set of logic rules $\mathcal{F}$, where each $f \in \mathcal{F}$ is a definite Horn clause of the form

$$f : \quad P_0(\mathbf{x}) \leftarrow P_1(\mathbf{x}) \wedge P_2(\mathbf{x}) \wedge \cdots \wedge P_k(\mathbf{x}), \quad \mathbf{x} \in \mathcal{X},$$

with head $P_0 \in \mathcal{P}$ and body predicates $P_1, \ldots, P_k \in \mathcal{P}$. For a fixed history $\mathbf{x}$, such a rule licenses the conclusion $P_0(\mathbf{x}) = 1$ whenever all body predicates satisfy $P_j(\mathbf{x}) =$

1. Each rule therefore specifies how new *truths* about $\mathbf{x}$ (previously unobserved predicate values) can be inferred from those already known.

For a fixed history $\mathbf{x}$, let

$$\Gamma_0(\mathbf{x}) := \{P \in \mathcal{P} : P(\mathbf{x}) = 1\}$$

be the set of *grounded predicates*, i.e., predicates that are known to hold directly from the data. **Forward chaining** iteratively enlarges this set of true predicates by applying rules whose bodies are already satisfied:

$$\begin{aligned}
\Gamma_{h+1}(\mathbf{x}) = \Gamma_h(\mathbf{x}) \\
\cup \Big\{ P_0 \in \mathcal{P} \;\Big|\; \big(P_0 \leftarrow P_1 \wedge \cdots \wedge P_c\big) \in \mathcal{F}, \\
P_1, \ldots, P_c \in \Gamma_h(\mathbf{x}) \Big\},
\end{aligned} \quad (3)$$

for $h = 0, 1, 2, \ldots$, until convergence. Since $\mathcal{P}$ is finite, this process terminates after finitely many steps, yielding a fixed point

$$\Gamma^\star(\mathbf{x}) := \Gamma_H(\mathbf{x}) \quad \text{with} \quad \Gamma_{H+1}(\mathbf{x}) = \Gamma_H(\mathbf{x}). \quad (4)$$

We refer to $\Gamma^\star(\mathbf{x})$ as the *logical closure* (or *logic-complete* set of predicates) for $\mathbf{x}$ with respect to $\mathcal{F}$: it contains exactly those predicates that are either grounded or derivable from grounded ones by finitely many rule applications, and no further rule can fire.

In FC-TPP, this symbolic picture serves as the conceptual target for the latent state: we will introduce a neural relaxation of forward chaining that operates on continuous predicate activations and event histories, aiming to approximate such logic-complete states in a form amenable to gradient-based learning.

# 3. Forward-Chaining Latent Temporal Point Processes (FC-TPP)

We consider a marked temporal point process on $\mathbb{R}_+ \times \mathcal{M}$. As in Sec. 2.1, we denote the history up to time $t$ by $\mathbf{x}_t = \{(t_j, m_j) \mid t_j < t\}$, which serves as the input for updating the latent state. Let $\Delta t_i = t_i - t_{i-1}$ with $t_0 := 0$. FC-TPP generates events sequentially in an autoregressive manner, but each conditional distribution over the next event's time and mark is governed by an explicit *latent symbolic state* that evolves via forward chaining.

## 3.1. Latent Symbolic State

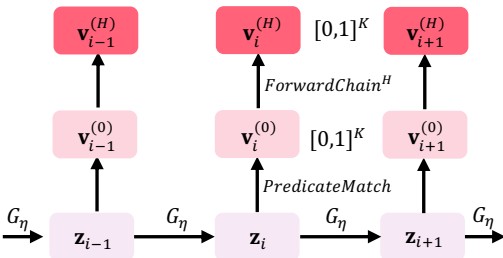

*Figure 2.* Latent state update in FC-TPP at event index $i$. The recurrent unit $G_\eta$ updates the history summary $\mathbf{z}_i$. Given $\mathbf{z}_i$, PredicateMatch produces initial predicate scores $\mathbf{v}_i^{(0)}$, which are refined by $H$-hop ForwardChain into $\mathbf{v}_i^{(H)}$. Vertical arrows above $\mathbf{z}_{i-1}$ and $\mathbf{z}_{i+1}$ indicate that the same mapping is applied at each step.

FC-TPP maintains two coupled latent representations, as shown in Fig. 2. First, a lower-level recurrent hidden state $\mathbf{z}_t$ summarizes the raw event history up to time $t$. Second, a higher predicate-level *latent symbolic state*

$$\mathbf{v}_t^{(H)} \in [0, 1]^K$$

captures high-level conditions over a fixed predicate vocabulary $\mathcal{P} = \{P_1, \ldots, P_K\}$, where $[\mathbf{v}_t^{(H)}]_k$ is the current activation (belief) of predicate $P_k$. At event times we write $\mathbf{z}_i := \mathbf{z}_{t_i}$ and $\mathbf{v}_i^{(H)} := \mathbf{v}_{t_i}^{(H)}$.

The recurrent state $\mathbf{z}_i$ is updated incrementally as new events arrive. Given the previous state $\mathbf{z}_{i-1}$, the inter-event gap $\Delta t_i$, and the new event mark $m_i$, we apply a GRU-style update:

$$\mathbf{z}_i = G_\eta(\mathbf{z}_{i-1}, \Delta t_i, m_i), \quad (5)$$

where $G_\eta$ is a lightweight neural recurrent unit parameterized by $\eta$.

From $\mathbf{z}_i$ we then construct the predicate-level state in two steps. First, we obtain *initial predicate scores* $\mathbf{v}_i^{(0)} \in [0, 1]^K$ by matching $\mathbf{z}_i$ to anchored predicate embeddings and then perform the $H$-step forward chaining (Sec. 3.2 for the descriptions of the operators):

$$\mathbf{v}_i^{(0)} = \text{PredicateMatch}(\mathbf{z}_i). \quad (6)$$

$$\mathbf{v}_i^{(H)} = \text{ForwardChain}^H(\mathbf{v}_i^{(0)}; \Theta_\mathcal{F}), \quad (7)$$

where $\Theta_\mathcal{F}$ is a learnable, embedding-based representation of a set of soft logic rules.

Intuitively, $G_\eta$ absorbs *local event evidence* into $\mathbf{z}_i$, PredicateMatch projects that evidence into predicate space to form a higher-level symbolic state $\mathbf{v}_i^{(0)}$, and ForwardChain$^H$ propagates rule implications to produce a *logically refined*, approximately rule-closed state $\mathbf{v}_i^{(H)}$. Together, $\eta$ and $\Theta_\mathcal{F}$ are the learned parameters governing the latent-state dynamics.

## 3.2. Neuro-Symbolic Reasoning Layer

The neuro-symbolic reasoning layer consists of two operators: $\text{PredicateMatch}(\cdot)$, and $\text{ForwardChain}^H(\cdot)$. We elaborate on each of them next.

**Predicate Matching:** We define the operator $\mathbf{v}_i^{(0)} = \text{PredicateMatch}(\mathbf{z}_i)$, which maps the recurrent state $\mathbf{z}_i$ to initial predicate activations $\mathbf{v}_i^{(0)}$. Specifically, for every $P \in \mathcal{P}$ we compute

$$\mathbf{u}_{i,P} = g_P(\mathbf{z}_i) \in \mathbb{R}^d,$$

using small MLPs $\{g_P\}$ or a shared projection with predicate-specific heads. Each predicate $P$ has a fixed *anchored* semantic embedding that forms a set $\mathbf{A} = \{\mathbf{a}_P : P \in \mathcal{P}\}$ (see Sec. 4.3 for how to establish the set $\mathbf{A}$).

We then measure how well the current event sequence (through $\mathbf{z}_i$) supports predicate $P$ using a cosine-based soft match between $\mathbf{u}_{i,P}$ and its anchor $\mathbf{a}_P$:

$$\mathbf{v}_i^{(0)} = [v_{i,1}^{(0)}, \ldots, v_{i,K}^{(0)}] \in [0, 1]^K,$$
$$v_{i,P}^{(0)} = \frac{\cos(\mathbf{u}_{i,P}, \mathbf{a}_P) + 1}{2}. \quad (8)$$

Here the cosine similarity is rescaled into $[0, 1]$, so $v_{i,P}^{(0)}$ can be interpreted as a soft confidence score for predicate $P$ at time step $i$ (*before* any rule-based reasoning).

**Forward Chaining:** We define the operator $\mathbf{v}^{(H)} = \text{ForwardChain}^H(\mathbf{v}^{(0)}; \Theta_\mathcal{F})$ (to simplify the notaion, we drop time step $i$ here), which recursively updates $\mathbf{v}^{(h)}$ at hop $h$ using the soft rule set $\Theta_\mathcal{F}$. One hop corresponds to applying all rules to perform the forward chaining once, propagating evidence from bodies to heads. Repeating this process $H$ times (until reaches the fixed point) yields $H$-hop reasoning, turning local predicate evidence in $\mathbf{v}_i^{(0)}$ into a multi-step, rule-consistent state $\mathbf{v}_i^{(H)}$.

We use smooth approximations of logical AND and OR. For

inputs $u_1, \ldots, u_m$, we define

$$\mathrm{softmin}_\tau(u_1, \ldots, u_m) := -\tau \log \sum_{j=1}^m \exp(-u_j/\tau), \quad (9)$$

$$\mathrm{softmax}_\tau(u_1, \ldots, u_m) := \frac{\sum_{j=1}^m u_j \exp(u_j/\tau)}{\sum_{j=1}^m \exp(u_j/\tau)}, \quad (10)$$

which recover $\min$ and $\max$ as $\tau \downarrow 0$, while remaining smooth for finite $\tau$.

*Step 1: Rule Grounding.* Each learnable rule parameter $\Theta_f \in \Theta_\mathcal{F}$ encodes an abstract symbolic pattern

$$\Theta_f = [\theta_{P_0}, \underbrace{\theta_{P_1}, \ldots, \theta_{P_c}}_{:=\Theta_b}]$$

which is a concatenation of embeddings for its head and body slots. To apply it, we align each slot embedding in the body part of the rule $\theta \in \Theta_b$ with the closest predicate embedding in the anchored predicate set $A = \{\mathbf{a}_P : P \in \mathcal{P}\}$:

$$(\mathbf{a}^*, P^*) = \arg \max_{\mathbf{a}_P \in A} \cos(\mathbf{a}_P, \theta), \quad \theta \in \Theta_b. \quad (11)$$

Here $P^*(\theta)$ is the index of the best-matching anchored predicate, and $\mathbf{a}^*(\theta)$ is its embedding.

*Step 2: Body Aggregation (soft AND).* Evidence from the body predicates is aggregated into a rule score using $\mathrm{softmin}_\tau$:

$$u_f^{(h)} = \mathrm{softmin}_\tau \left( \{\cos(\mathbf{a}^*, \theta), v_{P^*}^{(h)}\}_{\theta \in \Theta_b} \right). \quad (12)$$

The rule activation $u_f^{(h)}$ is high only when all body predicates are sufficiently supported: both strong body activations and strong semantic alignment contribute to firing the rule.

*Step 3: Head Update (soft OR).* If a predicate $P$ is the head of multiple rules, their contributions are aggregated:

$$v_P^{(h+1)} = \mathrm{softmax}_\tau \left( \{u_f^{(h)} : f \in \mathcal{F}, \ \mathrm{head}(f) = P\} \right). \quad (13)$$

In this update, the head predicate $P$ is strongly activated whenever *any* of its associated rules achieves a high score $u_f^{(h)}$, while $\mathrm{softmax}_\tau$ provides a differentiable approximation to logical OR.

In summary, let $\mathbf{v}^{(h)}$ denote the predicate vector after hop $h$, the one hop of forward chaining is updating the symbolic latent states using Eqs. (11)-(13). $H$-hop reasoning is obtained by applying the operation recursively $H$ times.

**Choice of $H$**    Classical forward chaining iterates until convergence, producing a logically closed set of facts. Our differentiable operator is a continuous relaxation of this

idea: each hop adds one layer of inferred evidence, and $H$ controls the effective reasoning depth. Running to a fixed point would be expensive and can raise the gradient saturation issues for gradient-based training; instead, we fix $H$ to balance expressive multi-hop reasoning with optimization stability (see ablations in the experiments).

The resulting $\mathbf{v}^{(H)}$ serves as a logic-completed latent state that both enforces logical consistency and helps impute missing high-level information.

### 3.3. Latent Symbolic State-Driven TPP Decoder

The event decoder depends *only* on the latent symbolic state $\mathbf{v}_i^{(H)}$. The next-event distribution factorizes as

$$p_\psi(\Delta t_i, m_i \mid \mathbf{v}_{i-1}^{(H)}) = p_\psi(\Delta t_i \mid \mathbf{v}_{i-1}^{(H)}) \, p_\psi(m_i \mid \mathbf{v}_{i-1}^{(H)}). \quad (14)$$

We parameterize $p_\psi(\Delta t_i \mid \mathbf{v}_{i-1}^{(H)})$ using an exponential-family distribution with parameters output by a small network applied to $\mathbf{v}_{i-1}^{(H)}$, and $p_\psi(m_i \mid \mathbf{v}_{i-1}^{(H)})$ as a categorical distribution with softmax logits from $\mathbf{v}_{i-1}^{(H)}$.

## 4. Learning and Generation

### 4.1. Learning Objective

Given an observed marked sequence $\mathbf{x} = \{(t_i, m_i)\}_{i=1}^N$, we form inter-event gaps $\Delta t_i = t_i - t_{i-1}$ and run the recurrent and reasoning layers forward to obtain $\mathbf{v}_{i-1}^{(H)}$ for each step $i$. The parameters $(\eta, \Theta_\mathcal{F}, \psi)$ are learned by maximizing the autoregressive log-likelihood of the next-event factors in Eq. (14):

$$\mathcal{L}(\eta, \Theta_\mathcal{F}, \psi) = \sum_{i=1}^N \log p_\psi(\Delta t_i, m_i \mid \mathbf{v}_{i-1}^{(H)}). \quad (15)$$

Since $\mathbf{v}_{i-1}^{(H)}$ is a deterministic function of the history via $G_\eta$, PredicateMatch, and ForwardChain$^H$, gradients backpropagate through the recurrent unit and the neuro-symbolic reasoning layer, encouraging the learned logic rules and predicate activations to be predictive of future times and marks.

### 4.2. Autoregressive Generation.

At test time, FC-TPP defines an autoregressive generator for new TPP samples. Starting from an initial state $(\mathbf{z}_0, \mathbf{v}_0^{(H)})$ (e.g., zeros or derived from context), we iteratively: (i) sample $(\Delta t_i, m_i)$ from $p_\psi(\Delta t_i, m_i \mid \mathbf{v}_{i-1}^{(H)})$, (ii) set $t_i = t_{i-1} + \Delta t_i$ and append the event $(t_i, m_i)$, (iii) update the recurrent state via $\mathbf{z}_i = G_\eta(\mathbf{z}_{i-1}, \Delta t_i, m_i)$, (iv) recompute the symbolic state by $\mathbf{v}_i^{(0)} = \mathrm{PredicateMatch}(\mathbf{z}_i)$ and $\mathbf{v}_i^{(H)} = \mathrm{ForwardChain}^H(\mathbf{v}_i^{(0)}; \Theta_\mathcal{F})$. This procedure

is repeated until a time horizon or stopping criterion is reached. Because the decoder always conditions on the rule-completed state $\mathbf{v}_{i-1}^{(H)}$, the same learned logic rules constrain both training and generation, shaping the distribution of emitted event times and types in a structurally consistent way.

### 4.3. Symbolic Prior Bank (SPB): Anchored Predicate Set & Initial Rules

We initialize the anchored predicate set $\boldsymbol{A} = \{\mathbf{a}_P : P \in \mathcal{P}\}$ by encoding each predicate name (and short description) with a pretrained language model and projecting the resulting embedding into the model's latent dimension via a learned linear map.

Rule embeddings $\Theta_{\mathcal{F}}$ are initialized from either human-specified templates or LLM-generated candidates, but are always treated as trainable parameters and updated jointly with the recurrent and decoder components during learning.

In addition, we can use prompt-based techniques to iteratively query an LLM for new candidate rules during training, filtering and adding them to $\Theta_{\mathcal{F}}$ so as to enrich the rule set with higher-quality structures as more data are observed. The overall modeling framework is shown in Fig. 1.

## 5. Experiments

### 5.1. Experimental Setup

**Datasets** We evaluate FC-TPP on six event sequence datasets spanning synthetic, semi-synthetic, and real-world settings. For (semi-)synthetic datasets, ground-truth logic rules are provided, enabling oracle studies under known constraints. In contrast, real-world datasets do not expose annotated rules; logical structure is therefore automatically inferred from data and used as guidance during generation. (i) *Synthetic*. **Syn@5** and **Syn@10** consist of event sequences with 5 and 10 predicates, respectively, sampled from TLPP (Li et al., 2020b). (ii) *Semi-Synthetic*. **LogiC-ity** (Li et al., 2024) simulates multi-agent urban events governed by customizable first-order logic rules. (iii) *Real-World*. **MIMIC-IV** (Johnson et al., 2020) contains ICU patient records, from which sepsis-related event sequences are extracted following Saria (2018). **EPIC-Kitchens-100** (Damen et al., 2020) provides annotated kitchen action sequences (verbs) used to construct temporal event histories. **IKEA ASM** (Ben-Shabat et al., 2021) captures human actions during furniture assembly; we focus on TV-bench assembly sequences. All datasets are split into 80%/10%/10% for training, validation, and testing. Additional preprocessing details are provided in Appendix B.1.

**Baselines** We compare FC-TPP against SOTA baselines from two categories. (i) *Neural TPP models*. We con-

sider four representative neural TPP baselines—**THP** (Zuo et al., 2020), **AVAE** (Mehrasa et al., 2019b), **GNTPP** (Lin et al., 2022), and **UFM-TPP** (Shou, 2025). (ii) *LLM-based models*. To examine the ability of LLMs to generate event sequences, we design several LLM-based baselines. **SP (Simple-Prompt)** treats the LLM as a sequence sampler and directly prompts it to output complete event sequences in the form "$(t_1, \texttt{event}_1), (t_2, \texttt{event}_2), \dots$". **QA (Question-Answering)** adopts a step-wise generation protocol following EDGAR (Castricato et al., 2021), where the LLM recursively predicts the next event and its timestamp. Finally, **LAMP** (Shi et al., 2023) is adapted by replacing its original single-event proposer with a full-sequence candidate generator that produces an entire event segment in a single pass. See implementation details in Appendix B.2.

**Evaluation Metrics** We evaluate generation quality under two complementary regimes. (i) *In-Distribution Generation*—For *synthetic* and *semi-synthetic* datasets with oracles rules. We report **KL Divergence** and **QQ-RMSE** (Xiao et al., 2017) against the true event-time distributions, and **MMD (Maximum Mean Discrepancy)** (Gretton et al., 2012) between generated and real sequences. Moreover, we also consider **RV (Rule Violation Rate)**, directly quantifying logical errors of rule adherence. For *real-world* datasets without ground truth rules, we report proxy realism metrics, including **DS (Discriminator Score)** (Desai et al., 2021)—defined as ($\texttt{accuracy} - 0.5$) of a two-layer LSTM trained to distinguish real from generated sequences, where values closer to zero indicate higher realism—and **GPTScore** (Fu et al., 2024), which measures the likelihood assigned by a LLM to textualized event sequences. (ii) *Rule-Conditioned (Out-of-Distribution) Generation*—in zero-shot generation, we employ LLM-based judges to assess **R-Score (Rule Adherence)**—the extent to which generated sequences satisfy the provided rules—and **C-Score (Contextual Plausibility)**—the plausibility of sequences within the given domain context under those rules. Both scores lie in $(0, 1)$. Details are shown in Appendix B.3.

### 5.2. Results and Analysis

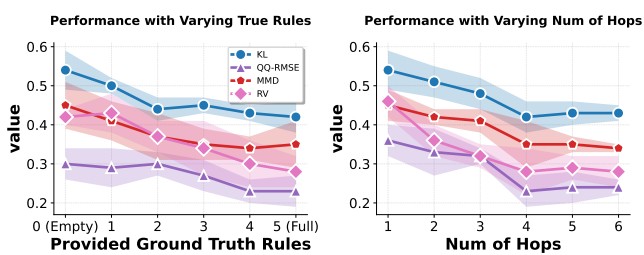

*Figure 3.* Key parameter selection on Syn@5. Left: provided oracle rules before training, from 0 (empty) to 5 (full). Right: the performance with varying forward chaining steps when full rules are provided.

*Table 1.* Rule scrambling test on Syn@5 dataset.

| Condition | Metrics | | | |
|---|---|---|---|---|
| | KL $\downarrow$ | QQ-RMSE $\downarrow$ | MMD $\downarrow$ | RV $\downarrow$ |
| True Rules | $\mathbf{0.42}_{\pm 0.04}$ | $\mathbf{0.23}_{\pm 0.04}$ | $\mathbf{0.35}_{\pm 0.06}$ | $\mathbf{0.28}_{\pm 0.06}$ |
| Scrambled Rules | $0.59_{\pm 0.07}$ | $0.36_{\pm 0.02}$ | $0.46_{\pm 0.03}$ | $0.71_{\pm 0.08}$ |
| No Rules | $0.54_{\pm 0.05}$ | $0.30_{\pm 0.04}$ | $0.45_{\pm 0.04}$ | $0.42_{\pm 0.02}$ |

### Analysis 1: Generation Performance and Comparison with Baselines

**Oracle Study with Known Rules**   We first conduct a controlled oracle study on synthetic data. As reported in Tab. 1, the rule scrambling test evaluates the robustness of rule-guided generation with mis-specified domain knowledge by preserving rule structure while breaking semantic correspondence. The results show that scrambled rules provide no benefit—and often substantially increase the rule violation rate (RV), indicating that the generated sequences frequently violate true underlying constraints—and even performing worse than using no rules at all. In contrast, only semantically correct rules consistently improve generation quality and reduce violations. Notably, even in the absence of any oracle rules, FC-TPP is able to learn effective rule structures from scratch, achieving satisfactory performance. This result provides an important validation of the model's ability to induce useful logic from data alone, thereby supporting its applicability to real-world settings where rules are unobserved.

With access to ground-truth logic rules on synthetic and semi-synthetic datasets, Tab. 2 shows that FC-TPP consistently outperforms most baselines across the evaluated metrics. Encouragingly, when reliable rules are available, explicitly incorporating logical guidance leads to substantial gains in generation quality.

**Rule-Guided Generation on Real-World Data**   Also in Tab. 2, on the real-world MIMIC-IV dataset, FC-TPP achieves statistically significant improvements over the strongest neural baseline (GNTPP), with relative gains of 11% in DS, 17% in GPTScore, and 14% in MMD. FC-TPP also attains superior performance on the IKEA ASM dataset. On EPIC-Kitchens-100, FC-TPP performs comparably to GNTPP, with only a small performance gap, suggesting competitive generation quality in more complex and visually grounded action domains.

### Analysis 2: Evidence of Parameter Efficiency, Data Efficiency, and Robustness   The synthetic setting allows for a controlled evaluation of parameter and data efficiency. By incorporating logical structure, FC-TPP attains strong generation performance with substantially fewer parameters than neural TPP baselines. As shown in Fig. 4, larger-capacity

models such as GNTPP require more parameters yet remain inferior to FC-TPP, highlighting the parameter efficiency induced by structured symbolic priors.

FC-TPP also exhibits good data efficiency. In few-shot settings on MIMIC-IV (see Analysis 5), training with only 40% of available data yields similar performance as the baselines trained on the full data, indicating that explicit rule-guided latent structure helps stabilize the generation performance.

The model also remains robust when rule priors are incomplete or unavailable. On Syn@5, FC-TPP successfully recovers meaningful rule patterns given partial rule sets, and even in the absence of oracle rules (no rules provided) achieves competitive performance (KL: 0.54, QQ-RMSE: 0.30, MMD: 0.45, RV: 0.42), surpassing most baselines (Tab. 1, and Fig. 3, left figure). Finally, ablations over the number of reasoning hops $H$ (Fig. 3, right figure) show that generation quality stabilizes around $H = 4$ when full oracle rules are provided, with larger values yielding only marginal gains. This suggests that modest multi-hop reasoning is sufficient to capture useful structure, demonstrating robustness to hyperparameter choices.

### Analysis 3: Exploration of Alternative Adaptive Reasoning Hop Schedule   In the current design, the reasoning hops $H$ is a small fixed hyperparameter (e.g., $\leq 10$), chosen based on validation performance. Empirically, we observe that performance saturates quickly with increasing $H$, indicating that shallow reasoning is sufficient in practice.

We also explored a dynamic stopping criterion, where reasoning continues until the rule-consistent state $\mathbf{v}^{(H)}$ convergence (i.e., by measuring $||\mathbf{v}^{(H+1)} - \mathbf{v}^{(H)}|| \leq \epsilon$). As shown in Tab. 13, Appendix. E.2, on MIMIC-IV, this yields slight improvements (MMD: $0.48 \rightarrow 0.45$) by adaptively adjusting reasoning depth, but at the cost of increased computation.

In practice, we find that fixed-hop design provides a trade-off. It achieves comparable performance while being simpler, more efficient, and more stable. Therefore, $H$ can be selected as a small constant, with dynamic variants as an available optional extension.

### Analysis 4: Ablation Study and Robustness Evaluations for LLM judges   The ablation results of MIMIC-IV dataset in Tab. 11 (Appendix D.6) demonstrate the contribution of each component in FC-TPP. Introducing neural pre-train module enabling finer-grained temporal reasoning and additional improvements, yielding gains of 17% in DS, 7% in GPTScore, and 4% in MMD, along with faster convergence and lower final training loss. The multi-step forward chaining similarly provides consistent gains across all evaluation metrics. The multi-round LLM rule modification and

*Table 2.* Comparison of different methods. Performance metrics are averaged across three different runs, which reported as (Mean ± SD). The best performance is in bold and also colored in purple.

| Methods | Synthetic Datasets | | | | | | | | Semi-Synthetic Datasets | | | |
|---|---|---|---|---|---|---|---|---|---|---|---|---|
| | Syn@5 | | | | Syn@10 | | | | LogiCity | | | |
| | KL ↓ | QQ-RMSE ↓ | MMD ↓ | RV ↓ | KL ↓ | QQ-RMSE ↓ | MMD ↓ | RV ↓ | KL ↓ | QQ-RMSE ↓ | MMD ↓ | RV ↓ |
| THP | $0.84_{\pm 0.06}$ | $0.45_{\pm 0.05}$ | $0.74_{\pm 0.08}$ | $0.48_{\pm 0.03}$ | $0.92_{\pm 0.08}$ | $0.76_{\pm 0.05}$ | $0.80_{\pm 0.05}$ | $0.63_{\pm 0.05}$ | $1.02_{\pm 0.05}$ | $0.87_{\pm 0.04}$ | $1.15_{\pm 0.10}$ | $0.70_{\pm 0.12}$ |
| AVAE | $0.67_{\pm 0.04}$ | $0.38_{\pm 0.03}$ | $0.62_{\pm 0.05}$ | $0.46_{\pm 0.01}$ | $0.79_{\pm 0.05}$ | $0.67_{\pm 0.04}$ | $0.81_{\pm 0.06}$ | $0.58_{\pm 0.05}$ | $0.86_{\pm 0.08}$ | $0.83_{\pm 0.09}$ | $1.05_{\pm 0.05}$ | $0.64_{\pm 0.06}$ |
| GNTPP | $0.62_{\pm 0.06}$ | $0.37_{\pm 0.03}$ | $0.54_{\pm 0.07}$ | $0.37_{\pm 0.03}$ | $0.71_{\pm 0.07}$ | $0.62_{\pm 0.03}$ | $0.76_{\pm 0.02}$ | $0.45_{\pm 0.04}$ | $0.80_{\pm 0.05}$ | $0.72_{\pm 0.04}$ | $0.94_{\pm 0.04}$ | $0.42_{\pm 0.06}$ |
| UFM-TPP | $0.54_{\pm 0.04}$ | $0.31_{\pm 0.02}$ | $0.48_{\pm 0.03}$ | $0.45_{\pm 0.04}$ | $0.72_{\pm 0.05}$ | $0.64_{\pm 0.06}$ | $0.70_{\pm 0.04}$ | $0.58_{\pm 0.09}$ | $0.75_{\pm 0.07}$ | $0.67_{\pm 0.05}$ | $0.88_{\pm 0.07}$ | $0.83_{\pm 0.04}$ |
| SP | $1.12_{\pm 0.07}$ | $0.62_{\pm 0.06}$ | $1.24_{\pm 0.11}$ | $0.64_{\pm 0.08}$ | $1.14_{\pm 0.08}$ | $0.78_{\pm 0.04}$ | $1.06_{\pm 0.10}$ | $0.75_{\pm 0.10}$ | $1.05_{\pm 0.09}$ | $0.79_{\pm 0.06}$ | $1.23_{\pm 0.08}$ | $0.78_{\pm 0.03}$ |
| QA | $0.78_{\pm 0.10}$ | $0.40_{\pm 0.05}$ | $0.71_{\pm 0.12}$ | $0.59_{\pm 0.08}$ | $0.85_{\pm 0.08}$ | $0.72_{\pm 0.06}$ | $0.87_{\pm 0.07}$ | $0.64_{\pm 0.05}$ | $0.96_{\pm 0.04}$ | $0.85_{\pm 0.08}$ | $1.16_{\pm 0.11}$ | $0.72_{\pm 0.07}$ |
| LAMP | $0.51_{\pm 0.03}$ | $0.29_{\pm 0.01}$ | $0.43_{\pm 0.02}$ | $0.41_{\pm 0.03}$ | $0.68_{\pm 0.02}$ | $0.56_{\pm 0.04}$ | $0.64_{\pm 0.05}$ | $0.53_{\pm 0.08}$ | $0.74_{\pm 0.06}$ | $0.70_{\pm 0.04}$ | $\mathbf{0.73_{\pm 0.05}}$ | $0.47_{\pm 0.00}$ |
| **Ours*** | $\mathbf{0.42_{\pm 0.04}}$ | $\mathbf{0.23_{\pm 0.04}}$ | $\mathbf{0.35_{\pm 0.06}}$ | $\mathbf{0.28_{\pm 0.06}}$ | $\mathbf{0.64_{\pm 0.06}}$ | $\mathbf{0.54_{\pm 0.12}}$ | $\mathbf{0.58_{\pm 0.07}}$ | $\mathbf{0.33_{\pm 0.00}}$ | $\mathbf{0.67_{\pm 0.06}}$ | $\mathbf{0.65_{\pm 0.07}}$ | $0.92_{\pm 0.10}$ | $\mathbf{0.31_{\pm 0.08}}$ |

| Methods | Real-World Datasets | | | | | | | | |
|---|---|---|---|---|---|---|---|---|---|
| | MIMIC-IV | | | EPIC-100 | | | IKEA ASM | | |
| | DS ↓ | GPTScore ↑ | MMD ↓ | DS ↓ | GPTScore ↑ | MMD ↓ | DS ↓ | GPTScore ↑ | MMD ↓ |
| THP | $0.46_{\pm 0.07}$ | $0.44_{\pm 0.03}$ | $0.78_{\pm 0.06}$ | $0.48_{\pm 0.05}$ | $0.50_{\pm 0.03}$ | $1.46_{\pm 0.08}$ | $0.48_{\pm 0.04}$ | $0.55_{\pm 0.03}$ | $0.82_{\pm 0.07}$ |
| AVAE | $0.46_{\pm 0.02}$ | $0.42_{\pm 0.03}$ | $0.72_{\pm 0.03}$ | $0.45_{\pm 0.06}$ | $0.51_{\pm 0.01}$ | $1.41_{\pm 0.09}$ | $0.48_{\pm 0.03}$ | $0.54_{\pm 0.02}$ | $0.75_{\pm 0.02}$ |
| GNTPP | $0.38_{\pm 0.03}$ | $0.60_{\pm 0.05}$ | $0.56_{\pm 0.04}$ | $\mathbf{0.37_{\pm 0.05}}$ | $\mathbf{0.62_{\pm 0.02}}$ | $1.29_{\pm 0.08}$ | $0.45_{\pm 0.03}$ | $0.58_{\pm 0.02}$ | $0.69_{\pm 0.04}$ |
| UFM-TPP | $0.44_{\pm 0.02}$ | $0.53_{\pm 0.06}$ | $0.65_{\pm 0.03}$ | $0.43_{\pm 0.04}$ | $0.52_{\pm 0.02}$ | $1.16_{\pm 0.08}$ | $0.46_{\pm 0.01}$ | $0.52_{\pm 0.02}$ | $0.71_{\pm 0.03}$ |
| SP | $0.45_{\pm 0.10}$ | $0.38_{\pm 0.07}$ | $0.94_{\pm 0.07}$ | $0.47_{\pm 0.05}$ | $0.41_{\pm 0.02}$ | $1.63_{\pm 0.10}$ | $0.48_{\pm 0.07}$ | $0.51_{\pm 0.04}$ | $1.38_{\pm 0.13}$ |
| QA | $0.42_{\pm 0.09}$ | $0.43_{\pm 0.05}$ | $0.71_{\pm 0.06}$ | $0.47_{\pm 0.03}$ | $0.46_{\pm 0.02}$ | $1.47_{\pm 0.14}$ | $0.47_{\pm 0.03}$ | $0.49_{\pm 0.05}$ | $0.79_{\pm 0.07}$ |
| LAMP | $0.41_{\pm 0.02}$ | $0.55_{\pm 0.04}$ | $0.58_{\pm 0.04}$ | $0.45_{\pm 0.02}$ | $0.54_{\pm 0.03}$ | $1.06_{\pm 0.09}$ | $0.39_{\pm 0.02}$ | $0.62_{\pm 0.03}$ | $0.64_{\pm 0.03}$ |
| **Ours*** | $\mathbf{0.34_{\pm 0.03}}$ | $\mathbf{0.70_{\pm 0.06}}$ | $\mathbf{0.48_{\pm 0.08}}$ | $0.43_{\pm 0.03}$ | $0.58_{\pm 0.02}$ | $\mathbf{0.95_{\pm 0.12}}$ | $\mathbf{0.34_{\pm 0.02}}$ | $\mathbf{0.65_{\pm 0.06}}$ | $\mathbf{0.62_{\pm 0.04}}$ |

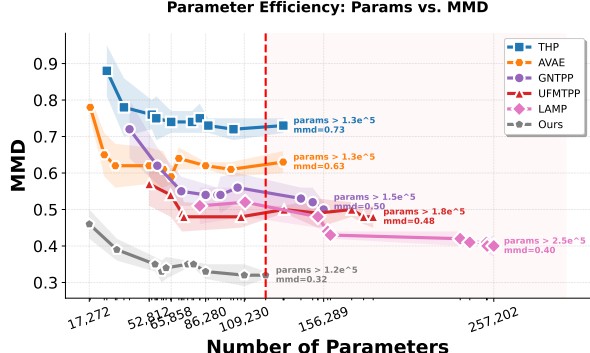

*Figure 4.* Parameter efficiency comparison of different methods on Syn@5 dataset. For each baseline, we varied the model complexity across 10 settings, from parsimonious to complex architectures.

querying module also improves model performance, though to a lesser extent than the first two modules.

We further evaluate robustness with respect to the choice of LLMs used for rule induction and evaluation. As shown in Tab. 4 (Appendix C), we consider models ranging from small-scale LLMs (Zhang et al., 2024; Team et al., 2024) to the OPT family (Zhang et al., 2022) and the GPT family (OpenAI, 2022). While rule quality generally improves with model scale—most notably between 1B and 10B parameters—even smaller LLMs achieve competitive performance. Moreover, as reported in Tabs. 5–6 (Appendix C), different LLM judges produce consistent scores with low variance, indicating that our conclusions are not sensitive

to the idiosyncrasies of any particular LLM. Also, we use different LLMs for rule extraction and evaluation, and extracted rules are not used verbatim—they are refined through trainable rule embeddings, therefore effectively mitigate circularity.

**Analysis 5: Few-Shot, Zero-Shot, and Partial-Data Generation** In many real-world applications—such as medical data anonymization and rare-disease modeling—event sequence generation likely operate under incomplete observations. We therefore evaluate FC-TPP in few-shot, zero-shot, and partial-data regimes.

Following the protocol of Li et al. (2020a), we assess few-shot generation by progressively reducing the proportion of training data. As shown in Fig. 5, FC-TPP consistently outperforms all baselines on MIMIC-IV across all data regimes. Performance degrades only marginally as data are reduced, and with just 40% of the training data, FC-TPP remains competitive with—or superior to—baselines trained on the full dataset. Moreover, by incorporating predefined medical knowledge into the neuro-symbolic layer, FC-TPP enables zero-shot generation of clinically meaningful sequences, achieving better generation quality than solely LLM-based baselines (Tab. 10, Appendix D.5).

To evaluate robustness under incomplete observations, we simulate realistic missing-event patterns by removing events from four temporal segments: initial ($[0, \frac{1T}{4})$), early-middle ($[\frac{1T}{4}, \frac{2T}{4})$), late-middle ($[\frac{2T}{4}, \frac{3T}{4})$), and final ($[\frac{3T}{4}, T]$). As shown in Fig. 6, FC-TPP outperforms all baselines in dif-

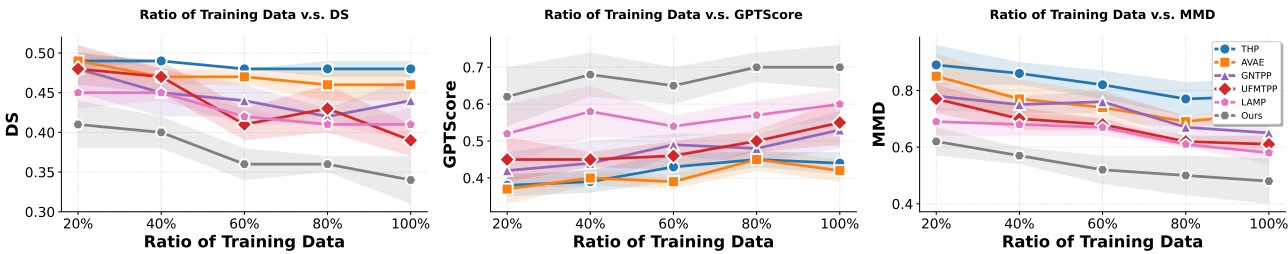

*Figure 5.* Few-Shot generation comparison for MIMIC-IV dataset, using MMD as evaluation metric. No comparison for SP and QA because they use pre-trained LLM.

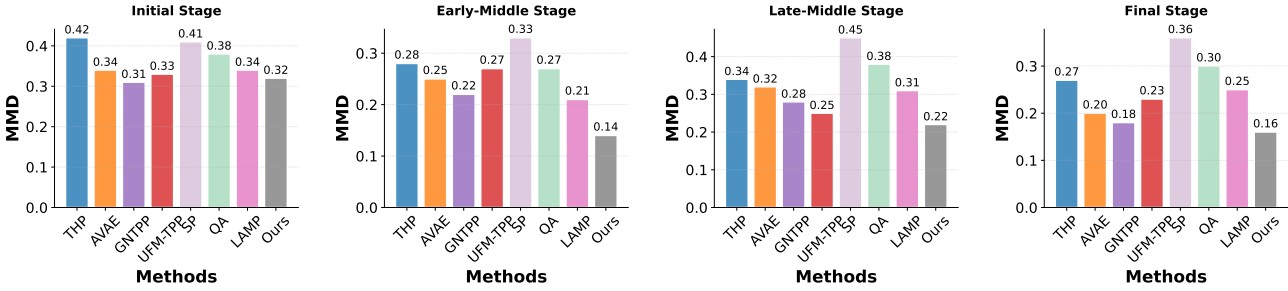

*Figure 6.* Generation performance comparison with different missing patterns for MIMIC-IV dataset. Define four missing data patterns over the time horizon: Initial Stage (missing events in the first $\left[0, \frac{1T}{4}\right]$), Early-Middle Stage ($\left[\frac{1T}{4}, \frac{2T}{4}\right]$), Late-Middle Stage ($\left[\frac{2T}{4}, \frac{3T}{4}\right]$), and Final Stage ($\left[\frac{3T}{4}, T\right]$).

ferent missingness regimes. Performance degradation is most pronounced when early events are missing—an expected limitation for autoregressive models that rely on initial context—but FC-TPP still maintains reasonable quality (MMD = 0.32). In contrast, it achieves the best performance when middle or late-stage events are missing, highlighting its robustness for generation from partial and incomplete histories.

## 6. Conclusion

We introduced FC-TPP, a neuro-symbolic framework that integrates TPPs with rule-guided reasoning for logic-consistent sequence generation. FC-TPP maintains an explicit predicate-based latent state and updates it via differentiable multi-hop forward chaining, so that the conditional distribution of the next event time and event type is driven by a reasoning-based state rather than a generic history embedding. Our approach enforces domain rules throughout the generative process while remaining trainable end-to-end. Empirically, FC-TPP achieves higher generation quality under data scarcity, stronger adherence to domain constraints or logics, and more controllable behavior than purely neural and prior neuro-symbolic baselines, indicating that explicit reasoning in latent space is a promising direction for TPP generation.

## Acknowledgements

This work was supported in part by the Key Program of the National Natural Science Foundation of China (NSFC) under Grant No. 72495131; the Shenzhen Stability Science Program 2023, Shenzhen Key Lab of Multi-Modal Cognitive Computing; the Shenzhen Science and Technology Program No. JCYJ20250604141038013; and the Longgang District Key Laboratory of Intelligent Digital Economy Security.

## Impact Statement

Our framework presents several positive societal impacts. By incorporating human-readable rules into neural generative models, it can produce interpretable synthetic sequences suitable for privacy-sensitive data sharing, rare-disease data augmentation, and debugging of high-stakes predictive models. However, potential risks emerge if generated data are deployed without rigorous validation: erroneous or biased rules may produce implausible or harmful samples, LLM-based evaluation could overestimate real-world reliability, and synthetic sequences might be misused for unverified clinical applications. To mitigate these concerns, future efforts should emphasize expert-involved rule curation, hybrid human–LLM evaluation protocols, full transparency of prompts and rule sets, and safeguards such as memorization checks and differential privacy in synthetic data release.

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

# Appendix

## A. Prompt Design

**Prompt for Querying Initial Rule Set**    The prompt for querying initial rule set is given below.

```
You are a logic reasoning assistant tasked with extracting high-level predicates and
logic rules based on observed temporal point process events.  Your goal is to identify
high-level predicates based on provided event types and logic rules that can capture
patient physical condition built on extracted predicates to identify the occurrence
patterns in temporal event sequences.  You are given a set of event types and some
of the examples of sequences.  The event type set consists of 38 different clinical
occurred events with clinical meaning.  And event sequences records in the format of
series of events, e.g.  (``event type'' @ ``event time'').

Your Goal:
Extract high-level predicates and generate reasonable and clinical meaningful logic
rules based on given clinical event types.

Instructions:
1.  Think aloud:  Propose reasonable number of high-level predicates.
2.  Think aloud:  Propose 3{5 candidate logic rules.
3.  Explain briefly why each rule might be effective.

Example Input:
Event type set:  [abnormal sysbp, abnormal spo2sao2, abnormal cvp, abnormal svr,
abnormal potassium meql, abnormal sodium, abnormal chloride, abnormal bun, abnormal
creatinine, abnormal crp, abnormal rbc count, abnormal wbc count, abnormal arterial
ph, abnormal arterial be, abnormal arterial lactate, abnormal hco, abnormal svo2 scvo2,
normal sysbp, normal spo2 sao2, normal cvp, normal svr, normal potassium meql, normal
sodium, normal chloride, normal bun, normal creatinine, normal crp, normal rbc count,
normal wbc count, normal arterial ph, normal arterial be, normal arterial lactate,
normal hco3, normal svo2 scvo2, low urine, intravenous fluids, use drugs, survival].

Event Sequences Examples:
Sequence 1:  [abnormal potassium meql @ 0.49, abnormal sodium @ 1.39, normal arterial
ph @ 1.78, ...]
Sequence 2:  [abnormal arterial be @ 1.58, normal arterial ph @ 1.58, abnormal sysbp @
1.62, ...]
Sequence 3:  [survival @ 0.00, abnormal potassium meql @ 0.01, normal arterial lactate
@ 0.01, ...]

Your Tasks:
Extract high-level predicates and generate symbolic rules based on these predicates.
For each rule, include 1{2 lines of explanation.

Output Example:
Extracted predicates:  A, B, C, ...
Generated rules:  IF A AND B AND C AND D, THEN E, ...
Explanation:  Because XXX, XXX...
```

**Prompt for Rule Set Refinement**    The prompt for refine initial/current rule set is given below.

```
You are a logic reasoning assistant tasked with extracting more logic rules except for
the given logic rules.  You are also given some observed temporal point process events
which cannot be explained well by the already provided logic rules.  Your goal is to
identify more different logic rules that can capture patient physical condition built
on previously extracted predicates to identify the occurrence patterns in the provided
temporal event sequences.  You are given some of the sequences that cannot be explained
well by the provided logic rules.  And event sequences records in the format of series
of events, e.g.  (``event time'' @ ``event time'').  Please note that the rules should
```

```
build on previous extracted high-level concepts

Your Goal:
Generate extra reasonable and clinical meaningful logic rules based on previously
extracted high-level predicate and provided sequences which cannot be explained well
by already provided logic rules.

Instructions:
1.  Think aloud:  Propose 1{2 more candidate logic rules.
2.  Explain briefly why each rule might be effective.

Event Sequences Examples that cannot Be Explained Well by the Provided Logic Rules:
Sequence 1:  [abnormal sodium @ 0.01, abnormal chloride @ 0.01, abnormal bun @ 0.01,
...]
Sequence 2:  [abnormal potassium meql @ 0.05, normal arterial ph @ 0.05, abnormal spo2
sao2 @ 0.10, ...]
Sequence 3:  [survival @ 0.00, normal arterial ph @ 1.81, normal arterial be @ 1.81,
...]

Provided Logic Rules:
Rule-1:  IF Cardiovascular Instability AND Electrolyte Imbalance THEN Renal Dysfunction
Rule-2:  IF Respiratory Dysfunction AND Acid{Base Disturbance THEN Cardiovascular
Instability
Rule-3:  IF Electrolyte Imbalance AND Renal Dysfunction THEN Inflammatory / Immune
Response
Rule-4:  IF Cardiovascular Instability AND Treatment Intervention THEN Survival
Rule-5:  IF Respiratory Dysfunction AND Inflammatory / Immune Response THEN Acid{Base
Disturbance

Your Tasks:
Generate symbolic rules that are different with the given logic rules, which can best
explain the given temporal point process sequences.  For each rule, include 1{2 lines
of explanation.

Output Example:
Generated rules:  IF A AND B AND C AND D, THEN E.
Explanation:  Because XXX, XXX...
```

## B. Dataset, Baselines, and Metrics

### B.1. Dataset

We summarize the dataset statistics in Tab. 3 and provide the preprocessing details below.

**Synthetic Datasets** *(i)* **Syn@5**: We create sequences consisting of 5 event predicates, sampled from a prespecified Temporal Logic Point Process (TLPP) (Li et al., 2020b). Specifically, we employ multiple pre-defined logic rules along with their weights to construct the intensity function, and then apply thinning algorithms (Ogata, 1981) to generate new events. To evaluate the scalability of the proposed model, we have created five distinct groups, with the sample size varying from 1000 to 9000 (see Appendix. D.7). In the experiment of main context, we use the dataset with 5000 samples and an average sequence length of 20.56 events. *(ii)* **Syn@10**: We make the synthetic setting more challenging by extending the number of predicates to 10 and further increasing the number of underlying rules, thereby enlarging the complexity of the underlying logical space. This setup enables us to assess the model's capacity to cope with richer and more intricate scenarios, as well as to evaluate its robustness and generalization ability under more demanding conditions. In our main experiments, we employ a dataset comprising 5000 samples with an average sequence length of 38.71 events per sample.

**Semi-Synthetic Datasets** *(iii)* **LogiCity** (Li et al., 2024): LogiCity is an urban-scale multi-agent simulator grounded in customizable first-order logic (FOL). It models diverse urban entities through semantic and spatial concepts, which are used to formulate FOL rules that govern agent behaviors. In our initial case study, we simulate 8 distinct agents (6 car agents and 2 pedestrian agent), each characterized by predicates such as `IsAtIntersection`, `Stop`, and `CollidingClose`, resulting in a total of 24 predicates. We generate 500 sequences to facilitate evaluation of interpretable rule-based reasoning

for generation tasks in dynamic environments. This platform serves as a testbed for neuro-symbolic methods in complex, interactive scenarios.

**Real-World Datasets** *(iv)* **MIMIC-IV** (Johnson et al., 2020): MIMIC-IV is a publicly available electronic health record dataset comprising patients admitted to the intensive care unit (ICU). We focus on those diagnosed with sepsis—a life-threatening condition and major cause of ICU mortality (Saria, 2018). We extract 2023 sequences with an average event count 33.10. Predicates are primarily categorized into two types: medication-related (e.g., `Use Drugs` or `Intravenous Fluids`) and clinical measurement-related (e.g., `Low Urine`, `Abnormal SpO2SaO2`, or `Abnormal ArterialPH`). These predicates support structured representation and logic-based reasoning for clinical event sequences. *(v)* **EPIC-100** (Damen et al., 2020; Song et al., 2024): This dataset is derived from a large-scale egocentric vision corpus, consisting of unstructured audio-visual recordings captured in natural home environments. It focuses on daily kitchen activities observed over multiple days. From the annotated action sequences, we extract temporal event histories comprising only cooking verbs—intentionally omitting the interacted objects to emphasize verb-centric reasoning. The dataset includes 68 verbs, such as `Put-In`, `Rinse`, `Put-On`, `Pour`, `Stir`, `Peel`, `Chop`, and `Slice`, among others. In total, the dataset contains 8656 sequences, with an average length of 34.84 events per sequence. *(vi)* **IKEA ASM** (Ben-Shabat et al., 2021): This dataset comprises 371 unique assembly configurations of four furniture types—side table, coffee table, TV bench, and drawer—each available in three colors: white, oak, and black. We focus specifically on the TV bench assembly task, which involves 18 action predicates such as `Pick Up Leg`, `Tighten Leg`, and `Attach Shelf to Table`. The subset includes 91 sequences with an average length of 24.46 events per sequence.

*Table 3.* Dataset statistics. For Epic-Kitchen-100 and IKEA ASM datasets, we scale the time horizon to 0-200.

| Category | Dataset | Statistics | | | |
|---|---|---|---|---|---|
| | | # Predicates | # Sequences | Events Average Length | Time Horizon |
| **Synthetic** | **Syn@5** | 5 | 5000 | 20.56 | 20 |
| | **Syn@10** | 10 | 5000 | 38.71 | 20 |
| **Semi-Synthetic** | **LogiCity** | 24 | 500 | 56.25 | 100 |
| **Real-World** | **MIMIC-IV** | 38 | 2023 | 33.10 | 118.42 |
| | **Epic-100** | 68 | 8656 | 34.84 | 1585 |
| | **IKEA ASM** | 18 | 91 | 24.46 | 5955 |

### B.2. Baselines

We choose state-of-the-art baselines considering two different fields:

**Neural Temporal Point Process Models** *(i)* **THP** (Zuo et al., 2020): THP models event sequences by parameterizing the conditional intensity function with a Transformer encoder that captures long-range temporal dependencies among past events. Event times and types are embedded and processed through self-attention, enabling flexible modeling of history-dependent dynamics without explicit parametric kernels. Event sequences are then generated by sampling from the learned intensity via standard point process simulation. *(ii)* **AVAE** (Mehrasa et al., 2019b): The model is a recurrent variational auto-encoder designed for modeling asynchronous action sequences. At each time step, the model utilizes the history of actions and inter-arrival times to generate a distribution over latent variables. A sample from this distribution is then decoded into probability distributions for the inter-arrival time and action label of the next action. To address the limitations of using a fixed prior in the traditional VAE framework, this model incorporates a prior net that enhances the learning process. *(iii)* **GNTPP** (Lin et al., 2022): The model is a comprehensive generative framework for neural temporal point process modeling. It leverages deep generative models as probabilistic decoders, including the temporal conditional diffusion denoising model, temporal conditional VAE, temporal conditional GAN, temporal conditional continuous normalizing flow, and temporal conditional noise score network models. The diverse combinations of encoders and decoders make the GNTPP highly flexible in approximating the target distribution of event occurrence times. For the encoder, the model incorporates both RNN-based approaches and self-attention mechanisms. In our experiments, we select the revised attentive history encoder and the VAE probabilistic decoder. *(iv)* **UFM-TPP** (Shou, 2025): It proposes a unified flow-matching framework for jointly modeling inter-event times and event types (marks), utilizing both continuous and discrete flow matching to achieve

integrated generation. This approach enables the model to directly map from simple noise distributions—such as exponential for times and uniform for types—to realistic event sequences, without relying on step-by-step recursion.

**LLM-Based Models** *(v)* **SP (Simple-Prompt LLM Generation)** (Si et al., 2022): For simple-prompt LLM generation, most existing works on prompt for event generation/extraction aim for teh event extraction (identifying event triggers and parameters from text); and story generation (generating narrative events (who did what)). In these works, "event" refers to events in the linguistic sense, rather than strictly stochastic modeling of temporal processes. Therefore, they are not directly equivalent to "TPP sequence generation". However, their methodologies can still inspire attempts at TPP generation: Prompt-as-sampler/Template prompting: Like the Simple Event Extraction Framework, this approach dispenses with complex probabilistic modeling and directly uses prompts to ask the LLM to output "$(t_1, \texttt{event type}_1), (t_2, \texttt{event type}_2)...$". *(vi)* **QA (QA-based LLM Generation)** (Castricato et al., 2021): For QA-based Generation, according to the idea of EDGAR (Castricato et al., 2021), the model can recursively generate the next event and its time through question-answering prompts. *(vii)* **LAMP** (Shi et al., 2023): A framework for TPPs that integrates a LLM. In particular, the LLM performs abductive reasoning to support the event sequence model. Guided by a few expert-annotated demonstrations, the LLM learns to suggest plausible causes for each candidate event. A search module then identifies previous events that align with these causes, and a scoring function evaluates whether the retrieved events could indeed lead to the proposed event. To adapt to TPP generation tasks, the original LAMP's proposer—which previously suggested candidate future events—has been replaced with an unconditional or conditional full-sequence candidate generator that produces a complete segment in a single pass. The process retains the LLM's abductive reasoning and re-ranking steps to filter and reorder candidates, ultimately outputting the final sample.

### B.3. Computation of Metrics

We evaluate generation quality under two complementary regimes:

**In-Distribution Generation** For fine-tuned generation, we can directly compare the generated sequences with the input sequences, since both are expected to follow similar patterns. Therefore, we can use following metrics to evaluate the quality of the generated sequences:

*(i)* **KL Divergence**: The synthetic datasets are generated by the temporal logic point process (TLPP). Therefore, we know the ground-truth logic rules and the corresponding rule weights, which allows us to compute the likelihood of each sequence. We first define the input sequence set and the output sequence set as $\boldsymbol{X} = \{\boldsymbol{X_i}\}_{i=1}^{n}$ and $\boldsymbol{Y} = \{\boldsymbol{Y_j}\}_{j=1}^{n}$. A sequence $\boldsymbol{S} = \{(t_l, a_l)\}_{l=1}^{N(\boldsymbol{S})}$ on the time horizon $[0, T]$. The event type $a_l \in \{1, ..., K\}$. Then the TLPP intensity for each event type $k$ can be computed as

$$\lambda_k(t|\mathcal{H}_t) = g(\eta_k(t)), \text{ where } \eta_k(t) = b_k + \sum_{f=1}^{F} w_{f,k}\phi_{f,k}(\mathcal{H}_t, t)$$

where $g(\cdot) > 0$, e.g., $g(x) = \exp(x)$. Total intensity can be written as

$$\Lambda(t) = \sum_{k=1}^{K} \lambda_k(t|\mathcal{H}_t)$$

Then compute the log-likelihood for each sequence. For an arbitrary sequence $\boldsymbol{S}$

$$l(\boldsymbol{S}) = \sum_{l=1}^{N(\boldsymbol{S})} \log \lambda_{a_l}(t_l|\mathcal{H}_{t_l}) - \int_0^T \Lambda(t)dt$$

To remove scale effects from sequence length or time window, we compute the per-event normalization and the per-time normalization.

$$l_{\text{per-event}}(\boldsymbol{S}) = \frac{l(\boldsymbol{S})}{N(\boldsymbol{S})} \text{ (if } N(\boldsymbol{S}) > 0), \ l_{\text{per-time}}(\boldsymbol{S}) = \frac{l(\boldsymbol{S})}{T}$$

Take standardization $l_{\text{per-event}}(\boldsymbol{S})/l_{\text{per-time}}(\boldsymbol{S})$ and collect vectors

$$\boldsymbol{l_X} = \{l(\boldsymbol{X}_1), ..., l(\boldsymbol{X}_n)\}, \ \boldsymbol{l_Y} = \{l(\boldsymbol{Y}_1), ..., l(\boldsymbol{Y}_n)\}$$

Estimate the log-likelihood density of two groups using kernel density estimation (KDE)

$$\hat{p}(x) = \frac{1}{nh} \sum_{i=1}^{n} k(\frac{x - l_X^{(i)}}{h}), \ \hat{q}(y) = \frac{1}{nh} \sum_{j=1}^{n} k(\frac{y - l_Y^{(j)}}{h})$$

Then the KL divergence between the two log-likelihood distribution are given by

$$\hat{\text{KL}}(\hat{p}||\hat{q}) = \frac{1}{n} \sum_{i=1}^{n} \log \frac{\hat{p}(l(\boldsymbol{X_i}))}{\hat{p}(l(\boldsymbol{Y_i}))}$$

*(ii)* **QQ-RMSE** (Xiao et al., 2017): Similarly we can get the KDE or histogram of $\boldsymbol{l_x}$ and $\boldsymbol{l_Y}$ on same axes. Then we compute the empirical quantiles $Q_X(\alpha), Q_y(\alpha)$ for $\alpha \in (0, 1)$. plot pairs $(Q_X(\alpha), Q_Y(\alpha))$. If distributions match, points lie on $y = x$. Implement discrete sampling by using $\alpha_k = \frac{k}{m+1}, k = 1, ..., m$. Then we can quantify the QQ deviation

$$\text{QQ-RMSE} = \sqrt{\frac{1}{m} \sum_{k=1}^{m} (Q_X(\alpha_k) - Q_Y(\alpha_k))^2}$$

*(iii)* **MMD (Maximum Mean Discrepancy)**: We computed MMD (Gretton et al., 2012) between the original sequences and the generated sequences after training. MMD quantifies the dissimilarity between the true data distribution $p^*(\boldsymbol{t})$ and the learned density $p(\boldsymbol{t})$ — lower is better. Now we provide the computation process:

Following above definition, the input sequence set is denoted as

$$\boldsymbol{X} = \{\boldsymbol{X_1}, \boldsymbol{X_2}, ..., \boldsymbol{X_n}\}, \text{ each } \boldsymbol{x_i} = [(t_{i1}, a_{i1}), (t_{i2}, a_{i2}), ..., (t_{iL_i}, a_{iL_i})]$$

And the output sequence set is denoted as

$$\boldsymbol{Y} = \{\boldsymbol{Y_1}, \boldsymbol{Y_2}, ..., \boldsymbol{Y_n}\}, \text{ each } \boldsymbol{y_j} = \left[(t_{j1}, a_{j1}), (t_{j2}, a_{j2}), ..., (t_{jL_j}, a_{jL_j})\right]$$

For each sequence in input set, we build sequence embedding

$$\boldsymbol{e_{ik}} = [\psi(t_{ik}); \text{emb}(a_{ik})] \in \mathbb{R}^d$$

where $\psi(\cdot)$ is the positional embedding for event time and $\text{emb}(\cdot)$ is the one-hot embedding for event type. Take the average of the entire sequence, we obtain

$$\tilde{\boldsymbol{X_i}} = \frac{1}{L_i} \sum_{k=1}^{L_i} \boldsymbol{e_{ik}}, \ \tilde{\boldsymbol{Y_j}} = \frac{1}{L_j} \sum_{k=1}^{L_j} \boldsymbol{e_{jk}}$$

Then, we define the kernel function (RBF kernel, i.e., Gaussian kernel)

$$k(\boldsymbol{A}, \boldsymbol{B}) = \exp\left(-\frac{||\boldsymbol{A} - \boldsymbol{B}||^2}{2\sigma^2}\right)$$

Finally, we can obtain unbiased MMD estimation

$$\text{MMD}^2(\boldsymbol{X}, \boldsymbol{Y}) = \frac{1}{n(n-1)} \sum_{i \neq j} k(\tilde{\boldsymbol{X_i}}, \tilde{\boldsymbol{X_j}}) + \frac{1}{n(n-1)} \sum_{i \neq j} k(\tilde{\boldsymbol{Y_i}}, \tilde{\boldsymbol{Y_j}}) - \frac{2}{n^2} \sum_{i=1}^{n} \sum_{j=1}^{n} k(\tilde{\boldsymbol{X_i}}, \tilde{\boldsymbol{Y_j}})$$

*(iv)* **RV (Rule Violation Rate)**: For the given rule set $\mathcal{F}$, the generated event sequences $\boldsymbol{X}$, we define rule trigger count as

$$T = \sum_{x \in \boldsymbol{X}} \sum_{f \in \mathcal{F}} \mathbb{I}\left[\bigwedge_{i=1}^{c} P_i(x) = 1\right]$$

Define rule violation count as

$$V = \sum_{x \in \boldsymbol{X}} \sum_{f \in \mathcal{F}} \mathbb{I} \left[ \bigwedge_{i=1}^{c} P_i(x) = 1 \wedge P_0(x) = 0 \right]$$

Then, the rule violation rate is defined as

$$RV = \frac{V}{T}$$

Here, we take an example to explain how to define the rule is obeyed. Consider a logic rule as $P_0 \leftarrow P_1 \wedge P_2$, namely $P_0$ must occur after $P_1$ and $P_2$ within $\Delta t$. Then, we can define

$$P_0(x) = 1 \Leftrightarrow \exists t_0 \in [\max(t_1, t_2), \max(t_1, t_2) + \Delta_t]$$

*(iv)* **DS (Discriminator Scores)** (Desai et al., 2021): The generated sequences are used to train the post-hoc sequence classifi-cation models (by optimizing a 2-layer LSTM) to distinguish between sequences from the original and generated sequences. First, each original sequence is labeled "*Real*", and each generated sequence is labeled "*Not Real*". Then, an off-the-shelf classifier is trained to distinguish between the two classes as a standard supervised task. Therefore, we obtain the discrimination scores (accuracy - 0.5). A score close to 0 is better, indicating the generated data is hard to distinguish from original data.

*(v)* **GPTScore** (Fu et al., 2024): A new framework that evaluates event sequence texts with generative pre-training models like GPT-3. It assumes that a generative pre-training model will assign a higher probability of high-quality generated event sequence text following a given instruction and context.

Note that the metrics KL and QQ-RMSE require ground truth from the temporal logic point process model. Consequently, they are only applicable to synthetic datasets.

**Rule-Conditioned (Out-of-Distribution) Generation**    However, in the case of zero-shot generation, the process is guided by modifying the underlying logic rules. As a result, the generated sequences may exhibit patterns that differ substantially from previously observed data, making direct comparison with input sequences less appropriate. To address this, we turn to LLMs as judges, evaluating the generated sequences along two dimensions: whether they adhere to the specified logic rules, and whether they align with realistic, domain-specific meaning.

*(vi)* **R-Score (Rule-based GPTScore)**: Measures how well the generated sequence adheres to the predefined logic rules, with the assessment performed by GPT. Higher scores indicate that the sequence better follows the specified logic rules.

*(vii)* **C-Score (Clinical GPTScore)**: Evaluates the overall plausibility and acceptability of the generated sequence, with the GPT assessing factors such as event order, timing, and domain knowledge. Higher scores indicate that the sequence is reasonable, meaningful, and consistent with expected patterns.

## C. Robustness Evaluations in LLM-based Rule Extraction and Evaluation

**Compare Different LLMs as Symbolic Prior Bank**    To compare the quality of rules provided by different LLMs, we consider TinyLlama-1.1B-Chat-v1.0 (Zhang et al., 2024), Gemma-2-2B-IT (Team et al., 2024), Opt family (Zhang et al., 2022) (Opt-125M, Opt-1.5B), and GPT family (OpenAI, 2022): (GPT-3.5, GPT-4o). As shown in Tab. 4, intriguingly, while the capability for logic rule generation generally improves with model scale—with a particularly notable leap from 1B to 10B parameters—we observe that even some smaller LLMs achieve competitive results. For instance, using OPT-1.5B as the rule generator still yields performance on MIMIC-IV that substantially outperforms the baselines in Tab. 2. This demonstrates the compatibility of our approach with a range of pre-trained language models.

**Compare Different LLMs as Judges**    We employed multiple LLM judges (including GPT-4o and smaller open-source models) to reduce dependency on a single model and reported results with means and variances. Across both in-distribution (Tab. 5) and zero-shot generation (Tab. 6) evaluations on MIMIC-IV dataset, we observe consistent performance with low variance between different LLMs. This indicates strong agreement in quality assessment across model scales, validating that our results are not reliant on the idiosyncrasies of any single LLM.

*Table 4.* Compare the effect of rules generated by different LLMs on MIMIC-IV dataset.

| Methods | Metrics | | | |
|---|---|---|---|---|
| | Loss ↓ | DS ↓ | GPTScore ↑ | MMD ↓ |
| **TinyLlama** | $162.94_{\pm 21}$ | $0.39_{\pm 0.00}$ | $0.65_{\pm 0.00}$ | $0.62_{\pm 0.02}$ |
| **Gemma-2** | $162.28_{\pm 27}$ | $0.36_{\pm 0.00}$ | $0.68_{\pm 0.04}$ | $0.54_{\pm 0.02}$ |
| **Opt-125M** | $163.21_{\pm 25}$ | $0.38_{\pm 0.01}$ | $0.63_{\pm 0.02}$ | $0.54_{\pm 0.02}$ |
| **Opt-1.5B** | $162.45_{\pm 24}$ | $0.34_{\pm 0.01}$ | $0.66_{\pm 0.02}$ | $0.50_{\pm 0.01}$ |
| **GPT-3.5** | $162.22_{\pm 16}$ | $0.34_{\pm 0.01}$ | $0.70_{\pm 0.03}$ | $0.50_{\pm 0.03}$ |
| **GPT-4o** | $\mathbf{161.33_{\pm 0.22}}$ | $\mathbf{0.34_{\pm 0.03}}$ | $\mathbf{0.70_{\pm 0.06}}$ | $\mathbf{0.48_{\pm 0.08}}$ |

*Table 5.* Robustness evaluation with multiple LLM judges on MIMIC-IV (GPTScore).

| Metrics | LLMs used | | | | |
|---|---|---|---|---|---|
| | GPT-4o | GPT-4o + Opt-1.5B | GPT-4o + Opt-1.5B + Opt-125M | GPT-4o + Opt-1.5B + Opt-125M + Gemma-2 | GPT-4o + Opt-1.5B + Opt-125M + Gemma-2 + TinyLlama |
| **GPTScore ↑** | $0.71_{\pm 0.00}$ | $0.71_{\pm 0.01}$ | $0.68_{\pm 0.01}$ | $0.68_{\pm 0.02}$ | $0.67_{\pm 0.02}$ |

*Table 6.* Zero-shot generation robustness on MIMIC-IV with multiple LLM judges (R/C-Scores). The pre-defined rules focus on drug effects (i.e., how medications influence clinical measurements).

| Metrics | LLMs used | | | | |
|---|---|---|---|---|---|
| | GPT-4o | GPT-4o + Opt-1.5B | GPT-4o + Opt-1.5B + Opt-125M | GPT-4o + Opt-1.5B + Opt-125M + Gemma-2 | GPT-4o + Opt-1.5B + Opt-125M + Gemma-2 + TinyLlama |
| **R-Score ↑** | $0.64_{\pm 0.00}$ | $0.64_{\pm 0.02}$ | $0.64_{\pm 0.02}$ | $0.63_{\pm 0.03}$ | $0.63_{\pm 0.02}$ |
| **C-Score ↑** | $0.65_{\pm 0.00}$ | $0.66_{\pm 0.02}$ | $0.67_{\pm 0.02}$ | $0.65_{\pm 0.02}$ | $0.65_{\pm 0.02}$ |

# D. Experimental Details

## D.1. Design of Rule Scrambling Test

To assess robustness to incorrect domain knowledge, we conduct a rule scrambling test, where we syntactically preserve the structure and frequency of rules while randomly permuting their semantic associations. This allows us to isolate whether performance gains stem from meaningful reasoning or merely from injecting additional structured bias. The rule template design follow the works of Li et al. (2020b; 2021); Yang et al. (2023); Kuang et al. (2024).

The scrambled rules preserve the syntactic structure and marginal predicate statistics of true rules, while breaking their semantic correspondence. We use two ways for scrambling:

- Head Permutation: For each ground truth rule, we fix the body predicates, randomly replace the head predicate (sampled from the same predicate pool, but not in the body). For example, if a ground truth rule is `RenalFailure ←Hypotension ∧ Sepsis`, the scrambed rule could be `Arrhythmia ←Hypotension ∧ Sepsis`. It remains rule arity and trigger frequency while break the causal/semantic consistency.
- Body Predicate Shuffle: For each ground truth rule, we fix the head predicate, randomly replace the body predicate (sampled from the same predicate pool, but not in the head). This is seen as a rule scrambling scheme that complements head permutation.

## D.2. Visualization of Input and Output Sequence Comparison

In the main text, we report quantitative analyses across multiple evaluation metrics. To complement these results, Fig. 7 presents a representative input–output comparison from the MIMIC-IV dataset after training convergence. The generated sequences basically align with the observed sequences in both event frequency and temporal structure, indicating that our model not only reconstructs the training data effectively but also achieves good-quality sequence generation.

## D.3. Representative Queried Rules with Real-World Significance on Real-World Dataset

We present part of representative logic rules queried from LLM for MIMIC-IV, EPIC-Kitchen-100, IKEA ASM real-world datasets in Tab. 7, Tab. 8, and Tab. 9. These logic rules are well-aligned with the domain common sense and realistic constraints of the datasets, and are therefore integrated as domain knowledge into the neuro-symbolic reasoning layer to support structured and interpretable inference.

*Table 7.* Part of logic rules queried from LLM for MIMIC-IV dataset.

**Part of Rules Queried from LLM for MIMIC-IV dataset**

**Rule-1**: `Renal Dysfunction ← Cardiovascular Instability ∧ Electrolyte Imbalance`
**Rule-2**: `Cardiovascular Instability ← Respiratory Dysfunction ∧ Acid{Base Disturbance`
**Rule-3**: `Inflammatory / Immune Response ← Electrolyte Imbalance ∧ Renal Dysfunction`
**Rule-4**: `Survival ← Cardiovascular Instability ∧ Treatment Intervention`
**Rule-5**: `Acid{Base Disturbance ← Respiratory Dysfunction ∧ Inflammatory / Immune Response`
...... ......

*Table 8.* Part of logic rules queried from LLM for Epic-Kitchen-100 dataset.

**Part of Rules Queried from LLM for Epic-Kitchen-100 dataset**

**Rule-1**: `Put-Into (Something) ← Open (Something) ∧ Take (Something) ∧ Cut (Something)`
**Rule-2**: `Pour-Into (Something) ← Turn-On (Something) ∧ Put (Something)`
**Rule-3**: `Turn-Off (Something) ← Turn-On (Something) ∧ Wash (Something)`
**Rule-4**: `Chop (Something) ← Take (Something) ∧ Cut (Something) ∧ Put-Down (Something)`
**Rule-5**: `Grab (Something) ← Open (Something) ∧ Take (Something) ∧ Close (Something)`
`∧ Put-Down (Something)`
...... ......

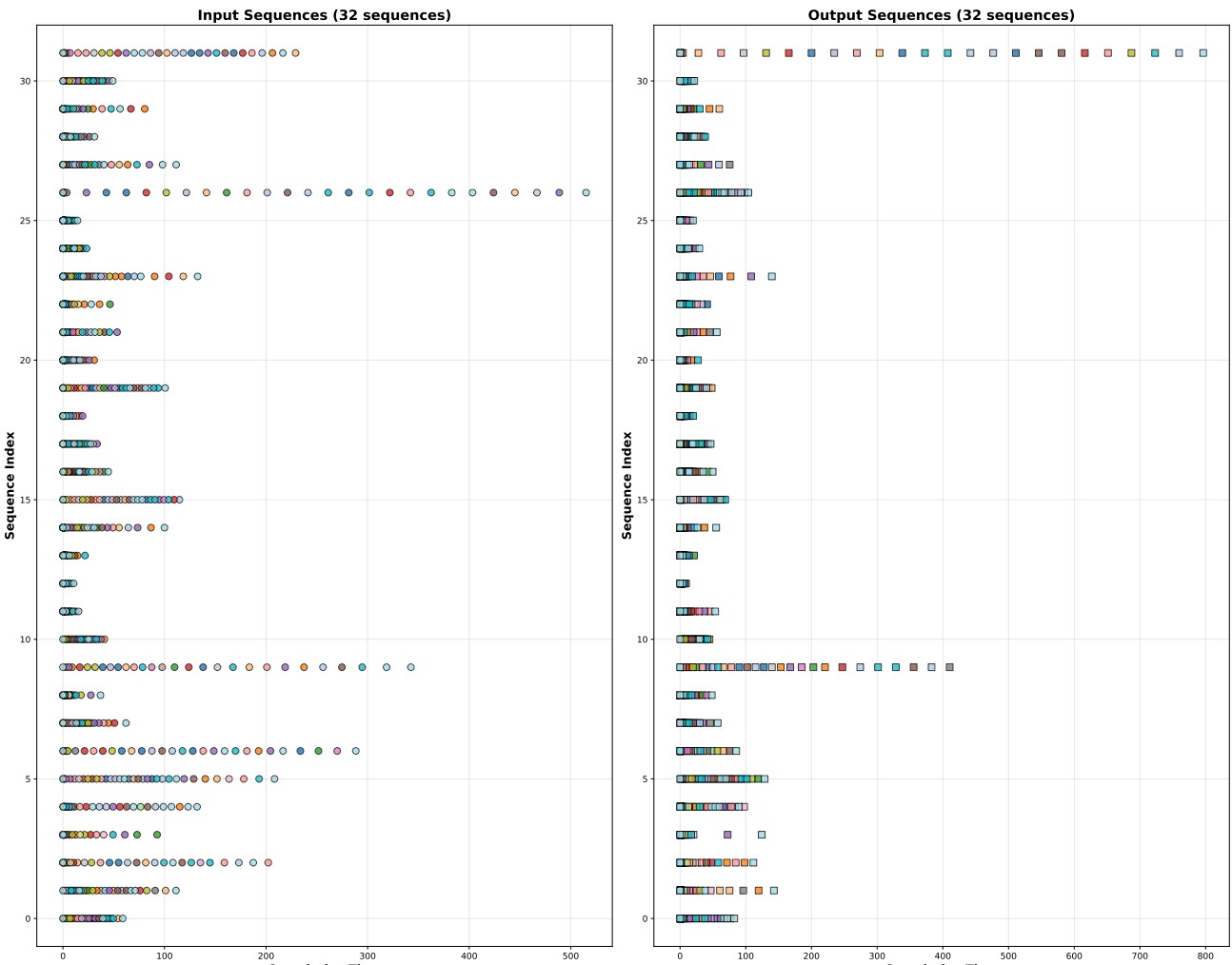

*Figure 7.* Comparison of model inputs and generated outputs on a representative batch from the MIMIC-IV dataset at convergence.

## D.4. Few-Shot Generation for Epic-Kitchen-100 dataset

As presented in Analysis 1 of the main text, our model exhibits slightly lower performance than the GNTPP baseline on the EPIC-Kitchen-100 dataset under full data (100% training) in terms of DS and GPTScore. However, results in Fig. 8 demonstrate stronger data efficiency when less training data is available: with 60% or fewer samples, our model outperforms most baselines on EPIC-Kitchen-100. In the extreme low-data regime using only 20% of the training data, our method substantially surpasses other approaches in sequence generation quality. Specifically, with 20% data, our model achieves DS = 0.45, GPTScore = 0.53, and MMD = 1.25. In comparison, the second-best results are achieved by GNTPP (DS = 0.46, GPTScore = 0.41) and by LAMP (MMD = 1.33). These findings underscore the suitability of our approach for few-shot generation, especially in data-scarce applications such as synthesizing rare disease data.

## D.5. Zero-Shot Generation for MIMIC-IV dataset

Encouragingly, the integration of domain knowledge into the neuro-symbolic reasoning layer enables zero-shot generation capability in our model. By pre-defining desired logic rules that the generated data must satisfy, our well-trained model can produce sequence data conforming to these rules in a zero-shot manner. As shown in Tab. 10, we designed logic rules focusing on two aspects: *drug effects (i.e., how medications influence clinical measurements)* and *lab test interactions (i.e., correlations among patient physiological indicators)*. Using an LLM as a judge, our method outperforms zero-shot baselines such as SP and QA across both R-Score and C-Score, demonstrating stronger generalization under novel rule constraints.

*Table 9.* Part of logic rules queried from LLM for IKEA ASM dataset.

**Part of Rules Queried from LLM for IKEA ASM dataset**

**Rule-1**: `Tighten Leg` ← `Pick Up Leg` ∧ `Align Leg Screw with Table Thread` ∧ `Spin Leg`
**Rule-2**: `Tighten Leg` ← `Align Leg Screw with Table Thread` ∧ `Spin Leg`
**Rule-3**: `Rotate Table` ← `Tighten Leg`
**Rule-4**: `Flip Table` ← `Tighten Leg`
**Rule-5**: `Attach Shelf to Table` ← `Flip Table` ∧ `Pick Up Shelf`
**Rule-6**: `Pick Up Leg` ← `Flip Table Top`
......            ......

*Table 10.* Zero-shot generation for MIMIC-IV dataset. We query LLM for pre-defined logic rule in two categories: drug effects (i.e., how medications influence clinical measurements), and lab test interactions (i.e., correlations among patient physiological indicators).

| Category | Rule Set | Metrics | |
| --- | --- | --- | --- |
| | | R-Score ↑ | C-Score ↑ |
| **Drugs** | **Rule-1**: `Treatment Intervention` ← `Cardiovascular Instability` ∧ `Respiratory Dysfunction`
**Rule-2**: `Treatment Intervention` ← `Electrolyte Imbalance` ∧ `Acid-Base Disturbance`
**Rule-3**: `Survival` ← `Treatment Intervention` ∧ `Renal Dysfunction`
......          ...... | 0.65 | 0.66 |
| **Lab Tests** | **Rule-1**: `Acid-Base Disturbance` ← ∧ `Cardiovascular Instability`
**Rule-2**: `Renal Dysfunction` ← `Respiratory Dysfunction` ∧ `Inflammatory/Immune Response` ∧
**Rule-3**: `Cardiovascular Instability` ← `Electrolyte Imbalance` ∧ `Acid-Base Disturbance`
......          ...... | 0.63 | 0.67 |

*Note*: **Drug**: For baseline **SP**: **R-Score**: 0.43, **C-Score**: 0.45. For baseline **QA**: **R-Score**: 0.46, **C-Score**: 0.45.
      **Lab Tests**: For baseline **SP**: **R-Score**: 0.51, **C-Score**: 0.55. For baseline **QA**: **R-Score**: 0.58, **C-Score**: 0.60.

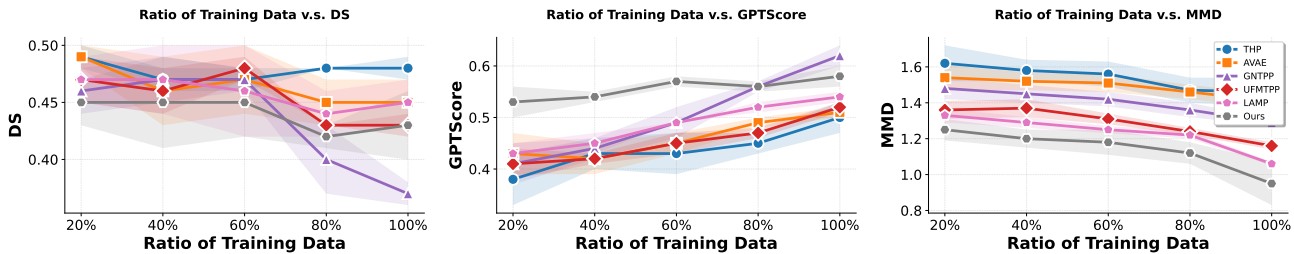

*Figure 8.* Few-Shot generation comparison for Epic-Kitchen-100 dataset, using MMD as evaluation metric. No comparison for SP and QA because they use pre-trained LLM.

## D.6. Ablation Study

*Table 11.* Ablation study on MIMIC-IV. We ablate the following modules: *(i)* Neural Pre-Train: if ablated, replace the GRU-style update to a simple MLP update, only take inter-event gap and the event mark as the input (without recurrent state). *(ii)* Forward Reasoning: if ablated, fix the forward reasoning step as 1. *(iii)* Rule Refine: if ablated, remove multi-round LLM rule refinement.

| Ablation | | | Metrics | | | |
|---|---|---|---|---|---|---|
| **Neural Pre-Train** | **Forward Reasoning** | **Rule Refine** | **Loss ↓** | **DS ↓** | **GPTScore ↑** | **MMD ↓** |
| ✗ | ✗ | ✗ | $173.08_{\pm0.16}$ | $0.48_{\pm0.03}$ | $0.59_{\pm0.03}$ | $0.57_{\pm0.04}$ |
| ✓ | ✗ | ✗ | $168.51_{\pm0.11}$ | $0.40_{\pm0.03}$ | $0.63_{\pm0.03}$ | $0.55_{\pm0.04}$ |
| ✓ | ✓ | ✗ | $162.34_{\pm0.28}$ | $0.37_{\pm0.02}$ | $0.68_{\pm0.02}$ | $0.51_{\pm0.03}$ |
| ✓ | ✓ | ✓ | $\mathbf{161.33_{\pm0.22}}$ | $\mathbf{0.34_{\pm0.03}}$ | $\mathbf{0.70_{\pm0.06}}$ | $\mathbf{0.48_{\pm0.08}}$ |

From the results of ablation study for MIMIC-IV dataset in Tab. 11, we found that introducing neural pre-train module enabling finer-grained temporal reasoning and additional improvements, yielding gains of 17% in DS, 7% in GPTScore, and 4% in MMD, along with faster convergence and lower final training loss. The multi-step forward chaining similarly provides consistent gains across all evaluation metrics. The multi-round LLM rule modification and querying module also improves model performance, though to a lesser extent than the first two modules.

## D.7. Scalability and Time Efficiency

To evaluate the time efficiency and scalability of our proposed method, we conduct experiments on the synthetic dataset Syn@5 with varying sample sizes: 1000, 3000, 5000, 7000, and 9000, with results shown in Fig. 9. Due to the incorporation of the neuro-symbolic reasoning layer and multi-round LLM rule refinement, our model exhibits slightly lower time efficiency compared to other deep neural methods. However, this does not constitute a substantial gap—training time per epoch remains comparable to baseline models across all dataset sizes.

Encouragingly, the additional complexity of our reasoning mechanism yields tangible benefits. Consistent with the findings in Analysis 3 in the main context and the results in Appendix. D.4, our model maintains strong performance in few-shot settings, which can also be found in Fig. 9. When the training sample size is reduced to only 1000 samples, our approach still outperforms most baseline models across the generative metrics. Moreover, performance improves steadily as sample size increases, demonstrating favorable scalability.

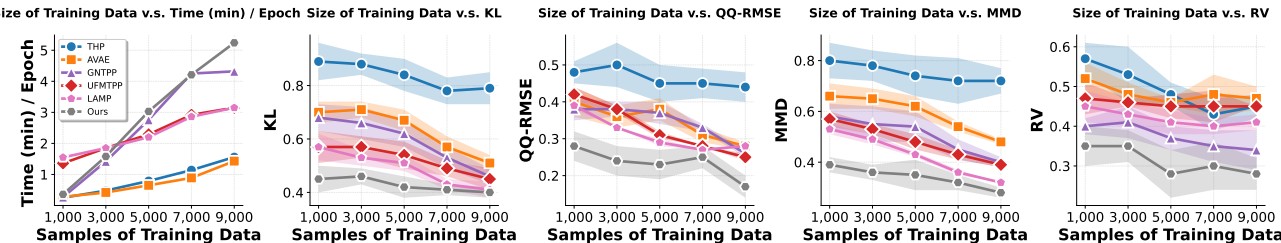

*Figure 9.* Time efficiency and scalability experiments for Syn@5 dataset. We vary sample size in $\{1000, 3000, 5000, 7000, 9000\}$.

# E. Additional Validation

## E.1. Evaluation using Standard TPP Metrics

We have also implemented evaluation on MIMIC-IV using the standard next-event prediction protocol, reporting time prediction error (MAE), event type prediction error rate (ER%), and event type prediction F1 Score. The results in Tab. 12 show that FC-TPP achieves improved predictive performance over strong neural TPP baselines. Beyond these metrics, FC-TPP additionally provides interpretable predicate states and rules, enabling reasoning-aware predictions, which standard TPP models lack.

*Table 12.* Evaluation using standard TPP metrics on MIMIC-IV dataset.

| Methods | Metrics | |
|---|---|---|
| | MAE $\downarrow$ | ER $\%\downarrow$ |
| THP | $3.16_{\pm 0.04}$ | $42.15_{\pm 3.62}$ |
| AVAE | $3.41_{\pm 0.12}$ | $48.33_{\pm 4.25}$ |
| GNTPP | $3.30_{\pm 0.28}$ | $45.72_{\pm 2.33}$ |
| UFM-TPP | $3.25_{\pm 0.17}$ | $42.83_{\pm 1.67}$ |
| LAMP | $3.14_{\pm 0.03}$ | $38.67_{\pm 2.33}$ |
| Ours* | $2.82_{\pm 0.10}$ | $37.50_{\pm 1.17}$ |

## E.2. Validation of Dynamic Reasoning Hop Schedule

In the current design, the reasoning hops $H$ is a small fixed hyperparameter (e.g., $\leq 10$), chosen based on validation performance. Empirically, we observe that performance saturates quickly with increasing $H$, indicating that shallow reasoning is sufficient in practice.

We also explored a dynamic stopping criterion, where reasoning continues until convergence (i.e., $||v^{(H+1)} - v^{(H)}|| \leq \epsilon$. As shown in Tab. 13, on MIMIC-IV, this yields slight improvements (MMD: $0.48 \to 0.45$) by adaptively adjusting reasoning depth, but at the cost of increased computation.

*Table 13.* Comparison of different reasoning hop schedules: dynamic v.s. fixed, using MIMIC-IV dataset.

| H Schedule | Metrics | | | | |
|---|---|---|---|---|---|
| | Avg. H | DS $\downarrow$ | GPTScore $\uparrow$ | MMD $\downarrow$ | Time Cost (mins) $\downarrow$ |
| Dynamic | $5.82_{\pm 0.33}$ | $0.32_{\pm 0.04}$ | $0.70_{\pm 0.00}$ | $0.45_{\pm 0.05}$ | $4.73_{\pm 0.02}$ |
| Fixed | $5.00_{\pm 0.00}$ | $0.34_{\pm 0.03}$ | $0.70_{\pm 0.06}$ | $0.48_{\pm 0.08}$ | $4.08_{\pm 0.01}$ |

In practice, we find that the fixed-hop design provides a trade-off. It achieves comparable performance while being simpler, more efficient, and more stable (e.g., avoiding vanishing gradients). Therefore, $H$ can be selected as a small constant, with dynamic variants as an available optional extension.

## E.3. Discussion on Whether to Use Raw Event History in the Decoder

The decoder is not independent of history. The event history is first encoded into the recurrent state $z_t$, and then mapped and refined into $v_t^{(H)}$. Thus, the decoder conditions on a history-informed latent state, rather than raw events.

Conditioning only on $v_t^{(H)}$ is intentional. It enforces a decoupling between raw history encoding and generation, so that event generation is driven by a reasoning-completed symbolic state rather than directly by noisy or long histories. This design improves controllability, as modifying predicates or rules leads to predictable changes in generation; directly conditioning on $z_t$ would re-entangle generation with raw history and weaken this control. Moreover, $v_t^{(H)}$ acts as a compact, structure-aware summary of history, filtering noise and capturing high-level dependencies, which is beneficial for long or partially observed sequences.

That said, the framework is flexible: the decoder can be extended to condition on both $v_t^{(H)}$ and $z_t$ if finer-grained temporal

information is needed. This modification is straightforward and does not affect the reasoning module.

We have designed additional experiments to directly address this beyond the conceptual discussion.

**Setup**    We generated synthetic event process by introducing some rare events driven by latent rule state $R_t = 0$ or $1$, defined via multievent combinatorial condition over a long time window (*not recoverable from short-term history*). Such as $R_t \leftarrow A \wedge B \wedge C$, where $A$, $B$, and $C$ are separated by long gaps. Background events follow a multivariate Hawkes process (strong local correlations). Target intensity:

$$\lambda_Y(t) = \mu_Y + \eta_R 1(R_t = 1) + \eta_H H_t$$

The synthetic event sequences contain 10 event types in total, including 2 rare event types. We use 5000 sequences, with an average of $\approx 33$ events per sequence. We consider: (1) FC-TPP (ours, symbolic $z_{\text{sym}}$ as state), (2) THP (history $h_{\text{hist}}$ as state), (3) Navie hybird (concat($z_{\text{sym}}$, $h_{\text{hist}}$)), and (4) Gated hybrid, where FC-TPP and THP produce separate predictions and the final output is selected via a simple confidence-based gating mechanism.

**Results**    The next-event prediction results (across all event types, with rare events accounting for approximately $13\%$ of true events) are shown in the table below:

*Table 14.* Comparison of different decoder architecture using synthetic dataset with rare events.

| Model | Metrics | |
| --- | --- | --- |
| | MAE $\downarrow$ | ER $\% \downarrow$ |
| **Ours\*** | $2.15_{\pm 0.04}$ | $17.34_{\pm 1.04}$ |
| **THP** | $2.57_{\pm 0.08}$ | $25.20_{\pm 2.20}$ |
| **Navie Hybrid** | $2.22_{\pm 0.08}$ | $17.02_{\pm 1.78}$ |
| **Gated Hybrid** | $2.08_{\pm 0.03}$ | $16.33_{\pm 1.56}$ |

The results in Tab. 14 show that (1) the Naive hybrid does not consistently improve performance in the above designed rare-event settings, and (2) the Gated hybrid achieves the best (slightly better than FC-TPP). This highlights that history information should be integrated in a controlled way rather than through direct fusion. Our FC-TPP retains a key advantage: it explicitly isolates a symbolic rule-based core for interpretability, while remaining readily extendable to hybrid designs. We will further elaborate on these extensions in the revised paper.

# F. Reproducibility Analysis

## F.1. Hyper-Parameter Selection

Our model is easy to implement and reproduce the results. We present the selected hyper-parameters on synthetic, semi-syntheti and real-world datasets in Tab. 15. The hyper-parameter selection metric is a trade-off between training converged loss, generation performance, and time efficiency.

## F.2. Computing Infrastructure

All synthetic data experiments and real-world data experiments, including the comparison experiments with baselines, are performed on Ubuntu 20.04.3 LTS system with Intel(R) Xeon(R) Gold 6248R CPU @ 3.00GHz, 227 Gigabyte memory.

## F.3. Computational and Memory Complexity

**Computational Complexity**    The forward pass complexity is dominated by the encoder and the reasoning layer:

- History encoder: a Transformer over $N$ events: $\mathcal{O}(N^2 \cdot d_e)$.
- Neuro-symbolic reasoning layer: the cost per sequence is $\mathcal{O}(|\mathcal{F}| \cdot H \cdot c)$, where $|\mathcal{F}|$ is the number of rules, $H$ is the number of reasoning hops, and $c$ is the average rule body size. This scales linearly with the key parameters.
- Event decoder: a feed-forward network applied at each of the $\hat{N}$ generated events: $\mathcal{O}(\hat{N})$.

*Table 15.* Descriptions and values of hyper-parameters used for models trained on the synthetic, semi-synthetic and real-world datasets.

| Hyper-Parameters | Value Used | | | | | |
|---|---|---|---|---|---|---|
| | **Syn@5** | **Syn@10** | **LogiCity** | **MIMIC-IV** | **EPIC-100** | **IKEA ASM** |
| Max Epochs | 32 | 32 | 50 | 50 | 64 | 64 |
| Max Reason Steps ($H$) | 3 | 3 | 7 | 5 | 5 | 5 |
| Batch Size | 32 | 32 | 32 | 32 | 64 | 32 |
| Predicate Embed Size | 32 | 32 | 64 | 32 | 32 | 32 |
| Hidden Size | 32 | 32 | 32 | 32 | 32 | 32 |
| # Layers | 2 | 2 | 2 | 2 | 2 | 2 |
| LLM Rule Extractor | – | – | – | GPT-4o | GPT-4o | GPT-4o |
| LLM Judge (Evaluation) | – | – | – | Opt-1.5B | Opt-1.5B | Opt-1.5B |
| Warm-up Learning Rate | False | False | True | True | True | True |
| Learning Rate | 1e-3 | 1e-3 | 1e-2 | 1e-2 | 1e-2 | 1e-3 |
| Optimizer | Adam | Adam | Adam | Adam | Adam | Adam |

**Memory Complexity**   The memory footprint is primarily determined by storing. The overall memory usage is linear in the sequence length and model dimensions:

- Sequence embeddings: $\mathcal{O}(N \cdot d_e)$ for the encoder outputs.
- Reasoning states: $\mathcal{O}(H \cdot K)$, where K is the number of predicates.

# G. Limitation and Broader Impacts

**Limitaion**   Our model relies on LLM-generated symbolic rules as priors, making its performance contingent on the quality, coverage, and prompt design of these rules. Biased or incomplete rules may adversely affect the generation process. Moreover, zero-shot evaluation depends on LLM-based judges, which raises concerns about circularity and the potential reinforcement of pre-existing biases. Finally, the quality of the generated data requires validation on high-quality rare-disease datasets—which are often difficult to acquire and typically require collaboration with medical institutions.

**Broader Impact**   Despite its limitations, our framework presents several positive societal impacts. By incorporating human-readable rules into neural generative models, it can produce interpretable synthetic sequences suitable for privacy-sensitive data sharing, rare-disease data augmentation, and debugging of high-stakes predictive models. However, potential risks emerge if generated data are deployed without rigorous validation: erroneous or biased rules may produce implausible or harmful samples, LLM-based evaluation could overestimate real-world reliability, and synthetic sequences might be misused for unverified clinical applications. To mitigate these concerns, future efforts should emphasize expert-involved rule curation, hybrid human–LLM evaluation protocols, full transparency of prompts and rule sets, and safeguards such as memorization checks and differential privacy in synthetic data release.

