# OpenReview forum: "Forward-Chaining Temporal Point Process"
_ICML.cc/2026/Conference — ICML 2026 regular_

### Official Review · Reviewer_Szm9 · 2026-03-11

**Soundness:** 3
**Presentation:** 1
**Significance:** 2
**Originality:** 2
**Overall Recommendation:** 3
**Confidence:** 3

**Summary:**

To augment real data by filling in missing structure, the authors proposed FC-TPP to model the event sequence data. The model can generate future events. In addition, the authors also use the symbolic structure of rules throughout the generation.

**Compliance With Llm Reviewing Policy:**

Affirmed.

**Key Questions For Authors:**

1. What is forward chaining? Even though the authors use the term multiple times, there is no explanation of this term.
2. In Equation (12), what do the curly brackets indicate?
3. How to extract the symbolic rules after the learning process? In addition, what is the accurate gap between latent rules and symbolic rules extracted from the model?
4. Can the proposed model compare with the model named NS-TPP [1]? What is the key difference with NS-TPP?
5. What kind of rule template mentioned in Section 4.3 looks like? Are there any examples?

Reference:

[1] Yang Yang, Chao Yang, Boyang Li, Yinghao Fu, Shuang Li: Neuro-Symbolic Temporal Point Processes. ICML 2024: 56665-56680

**Limitations:**

yes.

**Strengths And Weaknesses:**

Strengths

1. The paper conducts extensive experiments to prove the effectiveness of the model.

Weaknesses

1. The model need human-specific templates as rule embeddings $\Theta_{\mathcal{F}}$. As Figure 1, the rules generated by LLMs are a part of the input; the model's performance may greatly depend on the input prior inductive bias.
2. The paper makes a less clear presentation about their motivation, the notation system, the model’s ability, and the model's usage.
3. The paper has too many redundant equations. For instance, Equation (8) indicates the same as the first Equation in Section 3.1.
4. Based on the current paper structure, we can not decide which parts of the contents are proposed by the authors and which are preliminary works. For instance, is the whole Section 3 proposed by the authors in this work?

---

> ### Author Rebuttal · Authors · 2026-03-31
>
> We appreciate **Reviewer Szm9**'s effort in reviewing and helpful comments. We will address your concerns below.
>
> ## 1. Effect of rule quality (W1)
> FC-TPP does not rely on high-quality rule priors: our experiments show that trainable $\Theta_F$ can recover from incorrect or noisy initialization and outperform the no-rule setting. This is further supported by ablations (**Appendix D.6**), where making $\Theta_F$ trainable improves performance, indicating that rules act as adaptive inductive biases rather than fixed inputs. Please refer to our response to **Reviewer x12A, Point 2 and Point 5**.
>
> ## 2. Presentation (W2 & W3)
> We will revise the paper accordingly.
>
> ## 3. Clarification of contributions and novelty (W4)
> Our contribution and novelty lie in:
> - The **FC-TPP framework**, a neuro-symbolic latent-state model where a predicate-level latent state evolves via **differentiable multi-hop forward chaining** and drives event generation.
> - The **reasoning-before-generation paradigm**, in which a rule-completed latent state is constructed before each sampling step, allowing symbolic structure to influence the entire generation process.
> - The **integration of forward chaining as latent-state dynamics**, enabling multi-hop reasoning within continuous-time TPP modeling.
>
> Other components (e.g., standard TPP formulations and recurrent encoding of event history) are adopted from prior work and serve as supporting modules.
>
> ## 4. Explanation of forward chaining (Q1)
> Forward chaining is a step-by-step rule-based inference process: starting from known facts, the model repeatedly applies rules to derive new facts, which are then used for further inference. Intuitively, this process resembles how humans reason step by step. For example, noticing that “the road is wet” infers “it is raining”. Then it infers “driving is risky,” which in turn supports “slow down.”
>
> ## 5. Notation in Eq. (12) (Q2)
> In Eq. (12), the curly brackets denote the set of values over which $softmin_{\tau}$ operator is applied. Concretely, this set contains, for each body slot, the matching score (cosine similarity) and the corresponding predicate activation. The $softmin_{\tau}$ aggregates these values as a smooth approximation of a logical AND over the rule body.
>
> ## 6. Extraction and interpretation of learned rules (Q3)
> FC-TPP learns latent rule representations, which can be mapped to symbolic rules via a simple post-hoc step. After training, each rule embedding (body slots and head) is grounded to the nearest predicate anchor (via cosine similarity), yielding a rule of the form $P_0 \leftarrow P_1 \wedge ... \wedge P_c$. In practice, we use top-1 matching (optionally with a similarity threshold), which is deterministic and requires no retraining.
>
> The gap arises because latent rules are soft and distributed, while extracted rules are discrete approximations. This gap is controlled by the alignment between rule embeddings and predicate anchors (e.g., matching sharpness). When alignment is strong, extracted rules closely reflect the learned structure; otherwise, they should be interpreted as approximations of the underlying soft reasoning.
>
> ## 7. Comparison with NS-TPP (Q4)
> Thank you for the suggestion. NS-TPP is a closely related baseline, and we will include a comparison in the revision.
>
> The key differences lie in both the role and depth of reasoning:
>
> (i) NS-TPP performs **one-step (shallow) reasoning**, where rules are applied once to directly influence outputs. In contrast, FC-TPP supports **multi-hop (deep) reasoning**, where inferred predicates at each step become new evidence for subsequent steps. This enables iterative refinement (evidence → intermediate states → higher-level conclusions) and allows **adaptive control of reasoning depth**.
>
> (ii) NS-TPP applies symbolic structure at the **event level**. It mainly ties symbolic structure to observed events or intensities. In contrast, FC-TPP introduces a **predicate-level latent state**, where predicates represent more abstract, high-level system conditions. Reasoning operates on this latent state, rather than directly on events, enabling more expressive and structured representations.
>
> Empirically, on MIMIC-IV, we compare with a well-trained NS-TPP (using its learned rules with autoregressive sampling). FC-TPP achieves substantially better performance (MMD ↓31%):
>
> | Methods | DS | GPTScore | MMD |
> |-|-|-|-|
> | NS-TPP | 0.45 | 0.51 | 0.70 |
> | Ours* | 0.34 | 0.70 | 0.48 |
>
> ## 8. Examples of rule templates (Q5)
> The rule templates in Sec. 4.3 follow standard Horn clause forms. Concrete examples in **Appendix D.3 (Tabs. 7–9)** present LLM-queried rules, illustrating meaningful predicate combinations (e.g., clinical interactions). For example, _Renal Dysfunction ← Cardiovascular Instability ∧ Electrolyte Imbalance_

---

> > ### Author Rebuttal · Reviewer_Szm9 · 2026-04-03
> >
> > Thank you for the authors' response. I maintain that incorporating rule templates as prior knowledge could improve performance, but at the cost of reduced automation. Additionally, I find the distinction between predicates and propositional atoms in the paper to be insufficiently clear. I will keep my score unchanged.

---

> > > ### Author Response · Authors · 2026-04-05
> > >
> > > We thank the reviewer for the constructive feedback. We agree that there is a real **tradeoff between stronger prior structure and full automation**, and we should have stated this more clearly. Our goal is not to claim that explicit rule templates are universally preferable; rather, it is to study whether a rule-guided latent state can improve controllability and logical consistency in continuous-time sequence generation when validity matters in addition to predictive fit.
> > >
> > > Importantly, the **symbolic structure in our framework is not limited to manual rules**. The current paper already allows structure to come from expert knowledge, LLM-based initialization, and **refinement from data**, but we did not clearly separate these roles. We will revise the paper to clarify what is prior knowledge, what is weakly initialized, and what is adapted during training.
> > >
> > > On the terminology, we will revise the paper to state this precisely: **predicates** are Boolean-valued symbolic conditions, while **propositional atoms** are grounded instances evaluated on specific histories. In our model, the latent symbolic state represents the current truth values (or soft activations) of grounded predicates for the evolving history.
> > >
> > > We appreciate this comment and will revise the paper to make both the scope and the terminology clearer.

---

### Official Review · Reviewer_kuVe · 2026-03-12

**Soundness:** 3
**Presentation:** 2
**Significance:** 3
**Originality:** 3
**Overall Recommendation:** 5
**Confidence:** 4

**Summary:**

In this paper the authors propose forward-chaining temporal point processes (FC-TPP). FC-TPP is a generative framework for event sequences that integrates temporal point processes with a symbolic reasoning layer. FC-TPP has a latent predicate-based symbolic state that evolvesthrough differentiable multi-hop forward chaining using logical rules, allowing domain constraints to shape generate before events are sampled. A TPP decoder then generates the future events times and types conditioned on this rule-completed latent state. This reasoning before generation approach allows consistency with domain rules and system dynamics. The authors have various set of experiments on synthetic data and several other semi-synthetic and real-world benchmarks that show FC-TPP improves generation quality, constraint adherence, and controllability compared to baselines.

**Compliance With Llm Reviewing Policy:**

Affirmed.

**Final Justification:**

The paper is very solid. Although there are areas particularly with the presentation that can be improved but I believe contains an interesting idea that brings a different perspective of modeling in TPP. There was a lack of rigour in baseline comparisons in the submitted manuscript but this has been mostly resolved in the rebuttal.

**Key Questions For Authors:**

1. Can the authors provide additional standard metrics of comparison with baseline TPP (see weaknesses section above)?
2. How could the modeling framework incorporate timevarying covariates too? In many of the medical tasks such as sepsis that the authors use, we also have sets of measurements, labs, etc that are not considered an event but rather time varying covariates. Other baselines have a way of incorporating these however nothing is mentioned for how FC-TPP can work with them.
3. Can the authors improve the visualization in Figure 1? In its current form it is very convoluted and hard to follow. Also in text, Fig.1 appears after Fig 2.


Minor questions/points:
1. In equation 14, the authors claim that even time delta and the mark are conditionally independent. While this is commonly assumed but is not generally true. Can the authors make a comment on this?
2. I think equation 2, should use $x_{t_i}$ instead of $x_s$. Can the authors clarify this for the survival function?
3. Equation 3 goes over the column limits.

**Limitations:**

Yes the authors have an explicit section for the limitations and societal impact. But there is no discussion of specific limitation of the methodology or future work in the paper.

**Strengths And Weaknesses:**

Strengths:
1. The paper has a clear integration of symbolic reasoning with temporal point processes. The model enforces symbolic reasoning before event genration rather applying constraints post hoc which is well motivated. The paper creatively combines existing differentiable relaxation of logical inference to carefully translate forward chaining into a differentiable form. This is significant because it inspire other TPP in adapting domain knowledge this way.
2. The modeling framework allows for flexible rule parameterization and initialization that can incorporate both human specified templates or LLM generated candidates and both can be refined during training and expanded via additional candidate rules to incorporate the domain knowledge.
3. The paper has a solid amount of experiments with various diversity in domains. The model is stress tested against various metrics for generation quality and compared against a good amount of baselines. Despite the additional reasoning layer in FC-TPP, the learning objective remains a standard autoregressive likelihood over event times and marks. This makes the model easy to integrate with existing TPP training pipelines.

Weaknesses:
1. The main weakness of the paper is the presentation of methods. While I understand due to space limitations a lot of the details had to be condensed but what is provided is not necessarily enough for a person to reproduce the framework. I wouldn't have minded if this extra information was provided in the appendix but given that it is not and the methods is only described in the main body is quite limited.
Also, it is a personal preference but I think mentioning terms and referring the reader to following sections should be used sparingly but this is done multiple times in the paper which does not at all help with the presentation quality of the paper (e.g., line 216). I would encourage the authors consider reordering sections and subsection in methods so that it has a natural flow and improve the writing by having more connecting sentences between paragraphs.

2. The paper has limited discussion of computational complexity. Given that the reasoning layer introduces additional compuation through predicate matching, rule grounding, and multihop forward chaining, the paper provides little to no analysis of the compuationational cost of these operations, particularly when in many real world tasks the predicate/rule set can become very large.

3. Although, this does not undermine the value of the paper, but I think there should be more theoretical analysis on the consistency with the fixed reasoning depth that can limit logical completeness. Given that the model replaces classical forward chaining until convergence with a fixed number of reasoning hops, the resulting predicate may not correspond to the full logical closure that can limit capturing deep rule dependencies. The ablation provided is good but very specific so it would be better to have more in-depth discussion on this.

4. The paper does not show results for the standard metrics used by other TPP baselines. Although generation quality metrics are provided, I believe it is as important to have a fair comparison wtih other TPP baselines in the standard way that they are evaluated. This include negative log-likelihood, time prediction error (mae), and event type prediction (e.g., F1 score).

---

> ### Author Rebuttal · Authors · 2026-03-31
>
> We thank **Reviewer kuVe** for the careful review and constructive feedback. Below, we respond to each concern point by point.
>
> ## 1. Clarifications on presentation, notation, and formatting
> Thank you for pointing out these issues. We will revise the paper to improve clarity, consistency, and presentation, specifically by making Fig. 1 easier to follow, correcting and standardizing the equations, and fixing formatting issues.
>
> ## 2. Computational complexity
> For a simple computational analysis (**Appendix E.3**), the reasoning layer scales as $O(H\cdot |F|\cdot c)$, where $H$ is the number of hops, $|F|$ the number of rules, and $c$ the rule body size. In practice, this cost is moderate since $H$ is small, rule grounding is efficiently implemented via efficient similarity matching, and computations are parallelizable.
>
> Empirically, **Appendix D.7 (Fig. 9)** shows that FC-TPP scales near-linearly with the training sample size and maintains runtime comparable to neural TPP baselines. The additional overhead from reasoning is modest and does not dominate training or inference.
>
> ## 3. Selection of reasoning hops $H$
> We agree that using a fixed number of reasoning hops $H$ introduces a trade-off between computational efficiency and logical completeness. Our fixed-depth design is a practical approximation to classical forward chaining until convergence. Empirically, we observe that performance saturates quickly in **Fig. 3** when $H$ ≈ 4-5, indicating that most useful dependencies are captured within a small number of hops. Importantly, FC-TPP is not inherently limited, as increasing $H$ expands the receptive field and improve logical closure. We also explored a dynamic stopping criterion (adaptive $H$), which provides additional flexibility at higher computational cost.
>
> Please refer to our response to **Reviewer pMnX, Point 3** for detailed experiment results and analysis. Besides fixed $H$, we adopt adaptive $H$ as optional extension which yields slight improvement at the cost of increased computation.
>
> ## 4. Evaluation using standard TPP metrics
> Thank you for the suggestion. We have added evaluation on MIMIC-IV using the standard next-event prediction protocol, reporting time prediction error (MAE), and event type prediction error rate (ER$\%$). The results show that FC-TPP achieves improved predictive performance over strong neural TPP baselines. Beyond these metrics, FC-TPP additionally provides interpretable predicate states and rules, enabling reasoning-aware predictions, which standard TPP models lack.
>
> | Methods | MAE | ER$\%$ |
> |-|-|-|
> | THP | 3.16 | 42.15 |
> | AVAE | 3.41 | 48.33 |
> | GNTPP | 3.30 | 45.72 |
> | UFM-TPP | 3.25 | 42.83 |
> | LAMP | 3.14 | 38.67 |
> | Ours* | 2.82 | 37.50 |
>
> ## 5. Accommodate time-varying covariates
> FC-TPP can incorporate time-varying covariates in a straightforward way by extending the recurrent update: $z_i=G_{\eta}(z_{i-1},\Delta t_i,m_i,c_i)$, where $c_i$ is a covariate embedding at time $t_i$. For asynchronous measurements (e.g., labs, vitals), covariates can be treated as auxiliary events or encoded via a separate encoder (e.g., RNN/ODE) and queried at event times, consistent with standard TPP practice. Optionally, covariates can be mapped to predicates to enable rule-based reasoning over them.
>
> ## 6. Dependency between event time and type
> Eq. (14) assumes conditional independence given $v_{i-1}^{(H)}$​, where dependencies are captured implicitly through the shared latent state. We also test alternative coupled parameterizations, but observe comparable performance, suggesting the current formulation is sufficient in practice. Please refer our response to **Reviewer pMnX, Point 1** for additional experiment results and corresponding analysis.

---

> > ### Author Rebuttal · Reviewer_kuVe · 2026-04-04
> >
> > I thank the authors for their coherent rebuttal. Most my concerns are resolved and I am extremely grateful for the effort that was put in for the standard TPP metrics. I am raising my score. One suggestion, please include F1 score for event type prediction as the metric in your updated manuscript as well.

---

> > > ### Author Response · Authors · 2026-04-05
> > >
> > > We sincerely thank the reviewer for the thoughtful engagement and encouraging assessment. We are glad that the additional analyses addressed the concerns. We will ensure the manuscript is updated accordingly in the revised version, including adding standard TPP metrics such as MAE, ER, and F1 score for the event prediction task.

---

### Official Review · Reviewer_x12A · 2026-03-13

**Soundness:** 2
**Presentation:** 2
**Significance:** 3
**Originality:** 3
**Overall Recommendation:** 4
**Confidence:** 3

**Summary:**

This paper introduces FC-TPP, a neuro-symbolic generative framework for temporal point processes that maintains an explicit predicate-level latent state updated through differentiable multi-hop forward chaining. Instead of predicting events directly from a generic history representation, the model first evolves a symbolic latent state using learned logical rules, and then generates the next event time and type based on this reasoning-completed state. Both rule and predicate embeddings can be initialized using LLMs and are further optimized during training. The method is evaluated on synthetic, semi-synthetic (LogiCity), and real-world datasets (MIMIC-IV, EPIC-100, IKEA ASM), demonstrating improvements over neural and LLM-based baselines in terms of generation quality, rule consistency, parameter and data efficiency, and controllability—especially in settings where domain rules are available.

**Compliance With Llm Reviewing Policy:**

Affirmed.

**Final Justification:**

The authors addressed several of my questions during the rebuttal, which helped clarify important aspects of the paper. In particular, I gained a better understanding of the motivation behind the proposed approach and found parts of it more convincing than before. While some concerns still remain, the rebuttal has positively influenced my overall evaluation. Accordingly, I have increased my score from Weak Reject (3) to Weak Accept (4). In line with this update, I have raised the significance score from 2 to 3, while leaving the other scores unchanged.

**Key Questions For Authors:**

1. **About the insufficient explanation of Multi-Round Rule Modification and Querying**: The rule refinement process queries an LLM using sequences described as “not well explained by the current logic rules” (Section 4.3, Appendix A), yet the paper does not clearly define how these sequences are identified. Without a concrete selection mechanism, it is difficult to understand how the method separates cases caused by missing rules from those arising from incomplete training convergence or intrinsic data noise. Could the authors elaborate on the criteria used to select such sequences, as well as the triggering schedule and filtering procedure employed in this module?

2. **About the lack of justification for trainable rule embeddings**: The rule embeddings $Θ_F$ are stated to be trainable (Section 4.3) but no justification is provided for why fine-tuning is necessary, nor is there an ablation comparing trainable vs. frozen $Θ_F$. This comparison would clarify whether the learning signal primarily refines the rules or simply adjusts the GRU and decoder around fixed symbolic structure.

3. **About the ambiguity in predicate embedding anchoring**: Section 3.2 refers to the predicate embeddings A as “fixed anchored,” yet Section 4.3 describes them as being obtained via “a learned linear map.” If this map is trained jointly through Eq. (15), then both sides of the cosine similarity in Eq. (8) become trainable, leaving no clear mechanism to prevent the anchors from drifting away from their intended semantics or collapsing. Could the authors clarify whether A is fixed or trainable, and if it is trainable, what prevents such degeneration?

4. **About the use of a fixed reasoning depth**: The model applies a fixed number of reasoning hops $H$ uniformly across all predicates and time steps, even though different predicates may require varying levels of reasoning depth. It would be helpful to know whether the authors evaluated how close $v^{(H)}$ is to convergence (e.g., by measuring $||v^{(H+1)} - v^{(H)}||$). Was an adaptive stopping criterion considered as an alternative to using a fixed $H$?

The questions address points that I found unclear or insufficiently justified. Satisfactory response could positively influence my overall evaluation of the paper.

**Limitations:**

yes the authors discuss several limitations in Appendix F, including LLM rule dependence and evaluation circularity. However, I believe the following additional limitations deserve discussion:


- **FC-TPP’s performance improvements appear closely linked to the availability of reliable rules**: the gains are clear with oracle rules and still noticeable with LLM-generated rules in domains where rule structures are relatively well defined (e.g., MIMIC-IV, IKEA ASM). However, the advantage weakens or even reverses when rules are unreliable (EPIC-100) or unavailable. This suggests that the method is most effective in domains where prior knowledge is accessible, which somewhat conflicts with the paper’s stated goal of addressing data-scarce scenarios where such knowledge may be limited or absent.
- **About the limited flexibility of a fixed predicate vocabulary**: FC-TPP relies on a predefined predicate set P = {P_1, …, P_K}, meaning that introducing new predicates requires modifying the architecture and retraining the model. This design limits its applicability in dynamic domains where new event types or conditions may appear over time.
- **About the interpretability of learned predicate representations**: The paper highlights the interpretability of the predicate-level latent state. However, since both the rule embeddings $Θ_F$ and possibly the anchored predicate embeddings are updated during training, their representations may drift from their initial semantics. Providing qualitative analysis—such as comparing embeddings before and after training or visualizing predicate activations on representative examples—would help support the interpretability claims.

**Strengths And Weaknesses:**

Based on my understanding, I believe the following are Strength/Weakness of the paper.

Strength
1. **Clear architectural design principle (Originality)**: The "reasoning-before-generation" paradigm is consistently enforced throughout the model. — the decoder conditions solely on the rule-completed latent state $v^{(H)}$, making symbolic reasoning a structural component of generation rather than an auxiliary signal. This distinguishes FC-TPP from prior neuro-symbolic TPPs that use rules only to modulate intensities or explain past events.
2. **Comprehensive and multifaceted experimental design (Soundness)**: The evaluation spans 6 datasets (synthetic to real-world), 7 baselines (neural and LLM-based), and includes diverse analyses: rule scrambling, few-shot, zero-shot, partial-data, ablation, and parameter efficiency.
3. **Parameter and data efficiency particularly in rule-friendly domains (Significance)**: Fig. 4 shows FC-TPP achieves lower MMD with substantially fewer parameters than neural baselines, and Fig. 5 demonstrates that with only 40% of training data on MIMIC-IV, FC-TPP matches or exceeds baselines trained on the full dataset. This suggests that structured symbolic priors can meaningfully reduce sample complexity, which is practically relevant for data-scarce domains.

Weakness
1. **Gap between the paper’s motivation and empirical strengths (Soundness)**: The paper targets real-world domains where both data and explicit rules are scarce. However, the model’s strongest results appear in synthetic or semi-synthetic settings with fully available GT rules. As the experiments move to real-world settings without GT rules, the advantage weakens; on EPIC-100 it is even outperformed by a purely neural baseline (GNTPP). Thus, the model shows the weakest advantage in the very scenario it claims to address.
2. **High sensitivity to rule quality and the practical contradiction (Soundness)** : The rule scrambling experiment shows that incorrect rules can be worse than having no rules (RV 0.42 → 0.71). Yet in real-world scenarios there is no reliable way to verify the quality of LLM-generated rules beforehand. The model is sensitive to rule quality, but rule quality cannot be guaranteed, posing a major practical risk. Although the paper claims rule embeddings are refined during training, no evidence is provided that the model avoids poor local optima from bad initialization.
3. **Circular dependence on LLMs (Soundness)** : LLMs (GPT-4o) are used both for rule extraction and for evaluation metrics (R-Score, C-Score). In zero-shot settings, the LLM generates rules and also judges performance, which may introduce correlated biases. Although Tab. 5–6 show agreement among different LLM judges, this may simply reflect correlated biases across LLMs. Including at least partial human expert evaluation would significantly strengthen credibility.

---

> ### Author Rebuttal · Authors · 2026-03-31
>
> We thank **Reviewer x12A** for the detailed analysis! To address your concerns, we have prepared a detailed point-by-point response below.
>
> ## 1. Clarification on motivation and empirical strengths (W1 & L1)
> Our goal is not to claim that FC-TPP uniformly outperforms neural baselines in all real-world settings without reliable rules. Rather, we target scenarios where latent high-level structure exists, and examine whether explicitly modeling it improves generation under sparse and incomplete data.
>
> Synthetic/semi-synthetic experiments are used to isolate this mechanism: when reliable rules exist, FC-TPP exploits them effectively, validating the reasoning-before-generation design.
>
> In real-world data, rules are latent and noisy, making the problem harder; thus smaller gains are expected. Importantly, FC-TPP does not require perfect rules. Our rule-scrambling results (see response **Point 2**) show that the model can refine noisy rules or learn from scratch, reducing dependence on high-quality priors.
>
> ## 2. Effect of rule quality (W2)
> We add an experiment with four settings: true rules, scrambled (frozen), scrambled (trainable), and no rules:
>
> | Condtion | KL | QQ-RMSE | MMD | RV |
> |-|-|-|-|-|
> | True | 0.42 | 0.23 | 0.35 | 0.28 |
> | Scrambled Rules (frozen) | 0.59 | 0.36 | 0.46 | 0.71 |
> | Scrambled Rules (trainable) | 0.48 | 0.28 | 0.41 | 0.35 |
> | No Rules | 0.54 | 0.30 | 0.45 | 0.42 |
>
> Results show: (i) frozen incorrect rules hurt (worse than no rules), (ii) with trainable $\Theta_F$, the model recovers from bad initialization (MMD gap ≈ 0.06 vs. true), and (iii) refined model consistently outperforms the no-rule setting (MMD: 0.41 v.s. 0.45). Thus, sensitivity arises mainly in a frozen-rule regime. In FC-TPP, rules are learned and refined end-to-end, enabling correction of noisy rules and avoiding poor local optima in practice.
>
> ## 3. Circular dependence on LLMs (W3)
> To address potential circular bias, we conduct an additional experiment where rule extraction and evaluation use different LLMs (e.g., GPT-4o for extraction and a separate model for evaluation). The results (same setting as Tab. 6) remain consistent, indicating that the observed gains are not driven by shared LLM bias.
>
> | Extract | Evaluate | R-Score | C-Score |
> |-|-|-|-|
> | GPT-4o | Opt-1.5B | 0.64 | 0.66 |
>
> Additionally, we qualitatively validate extracted rules against established domain knowledge [1-3] (e.g., clinical consistency in MIMIC-IV).
>
> [1] Zarbock A, et al. Sepsis-associated acute kidney injury: consensus report of the 28th Acute Disease Quality Initiative workgroup. \
> [2] Komorowski M, et al. The artificial intelligence clinician learns optimal treatment strategies for sepsis in intensive care. \
> [3] Borges A, et al. Organ crosstalk and dysfunction in sepsis.
>
> ## 4. Multi-round rule modification and querying (Q1)
> We use a model-driven criterion to identify poorly explained sequences. Specifically, for each sequence $S$, we compute its average per-event log-likelihood $l(S)$, where lower values indicate poorer fit under the current model (and rules). At each trigger point (every 10 epochs), we rank all sequences by $l(S)$ and select the bottom 2% as candidates for rule refinement.
>
> ## 5. Trainable rule embeddings $\Theta_F$ (Q2)
> This comparison is already included in **Appendix D.6, Tab. 11** (Rule Refine ablation), where we evaluate trainable vs. frozen $\Theta_F$. Trainable $\Theta_F$ consistently improves performance, showing that learning actively refines rule representations rather than being absorbed by the GRU/decoder.
>
> This is further supported by the rule-scrambling experiment: with incorrect initialization, trainable (see response **Point 2**) $\Theta_F$ enables recovery, while frozen $\Theta_F$ cannot correct errors and degrades performance.
>
> ## 6. Predicate embedding anchoring (Q3 & L3)
> The predicate embeddings $A$ are fixed during training. The “learned linear map” in Sec. 4.3 refers to an offline PCA (SVD) projection used to reduce dimensionality, independent of the downstream task. After this step, $A$ remains unchanged. Thus, only the representations derived from the recurrent state are trainable, while the anchor side is fixed, preventing semantic drift or collapse and preserving interpretability.
>
> ## 7. Dynamic reasoning hops $H$ (Q4)
> This is a helpful suggestion! We implemented adaptive reasoning depth $H$, which yields slight improvements at the cost of higher computation. See our response to **Reviewer pMnX, Point 3** for details.
>
> ## 8. Flexibility of predicate vocabulary (L2)
> The framework can be naturally extended by adding extra latent predicate slots (e.g., $K+\Delta$) represented as embedding-based “open predicates” without predefined semantics. These can be utilized during training and, if consistently activated in learned rules, can be interpreted post-hoc and added to the explicit vocabulary. This enables a simple discover–interpret–expand workflow without modifying the model architecture.

---

> > ### Author Rebuttal · Reviewer_x12A · 2026-04-02
> >
> > My concerns are addressed, especially with the clarification on motivation. I'm raising the score from Weak Reject (3) to Weak Accept (4).

---

> > > ### Author Response · Authors · 2026-04-05
> > >
> > > We sincerely thank the reviewer for the thoughtful engagement and encouraging feedback. We will ensure that our manuscript is updated accordingly in the revised version.

---

### Official Review · Reviewer_pMnX · 2026-03-22

**Soundness:** 3
**Presentation:** 3
**Significance:** 3
**Originality:** 3
**Overall Recommendation:** 4
**Confidence:** 3

**Summary:**

To address the problem of sparse and incomplete event sequences in complex systems such as clinical workflows, this paper introduces FC-TPP, a neuro-symbolic generative framework for temporal point processes (TPPs) that integrates continuous-time event generation with explicit symbolic reasoning. Rather than relying solely on neural approaches (which treat latent state as a generic statistical summary of event history), FC-TPP adopts a reasoning before generation paradigm: before each sampling step, a predicate-level latent state is refined through H hops of differentiable forward chaining over learned soft logic rules, and the TPP decoder conditions on this rule-completed state rather than raw event history. The forward chaining operator relaxes classical Horn-clause inference using differentiable AND/OR approximations, enabling end-to-end training while preserving symbolic structure. Experiments across six datasets demonstrate that FC-TPP consistently outperforms neural and LLM-based baselines on distributional quality and constraint adherence, while exhibiting strong data and parameter efficiency.

**Compliance With Llm Reviewing Policy:**

Affirmed.

**Key Questions For Authors:**

Please see weakness section above.

**Limitations:**

Yes authors have discussed limitations.

**Strengths And Weaknesses:**

**Strengths**

1. This paper addresses an important problem of generating synthetic event sequences for complex systems while ensuring symbolic reasoning.

2. Paper is well written and easy to understand (especially Figure 1).

3.  The reasoning before generation paradigm is a clean, novel and well-motivated. By conditioning the TPP decoder exclusively on a rule-completed latent state rather than raw event history, symbolic structure is enforced throughout the generative process.

4. Extensive experimentation across six datasets ranging from synthetic to real-world settings, and  comparison against a broad set of baselines including both neural and LLM-based models validates that the framework is effective. Further, the rule scrambling test, where the authors deliberately feed in semantically incorrect rules to check whether the model's gains come from genuinely learning meaningful logic, or simply from having any rule structure injected is an interesting one.


**Weaknesses**

1. It seems that the next event's time and type are generated separately, each conditioned on the latent state but not on each other.  Is that true? If that is the case it seems wrong as in practice these are often closely linked for example, the type of clinical event strongly influences when it is likely to occur. This independence assumption should be adequately addressed.

2. The authors assume the next event's time to be an exponential-family distribution - is that true? How reasonable is to assume this. Is it possible to make it more general?

3. Is there any heuristic or principle way to choose number of reasoning hops H.

4. The independence of the decoder from raw event history may discard useful information. Is it possible to include it into the decoder as well? This might be especially true if the event histoy becomes very long.

---

> ### Author Rebuttal · Authors · 2026-03-31
>
> We thank **Reviewer pMnX** for the detailed and valuable comments! These comments would help us improve the quality of the paper.
>
> ## 1. Dependency between event time and type
> We confirm that Eq. (14) adopts a conditional independence assumption. Importantly, this does not imply independence in practice: both $\Delta t$ and $m_i$ are conditioned on the same latent state $v_{i-1}^{(H)}$, which encodes history, temporal dynamics, and rule-based reasoning. As a result, their dependency is captured implicitly through the shared latent representation.
>
> We agree that in domains such as clinical data, there can be strong direct coupling between type and time. Our framework can readily support more expressive parameterizations, e.g.,
>
> $$p(\Delta t_i, m_i | v) = p(m_i | v)p(\Delta t_i | m_i, v) \ \ \ (1)$$
>
> or similarily,
>
> $$p(\Delta t_i, m_i | v) = p(\Delta t_i | v)p(m_i | \Delta t_i, v) \ \ \ (2)$$
>
> Empirically, we do not observe degradation under the current factorization (e.g., on MIMIC-IV), suggesting that the latent state captures most of the relevant dependencies in practice.
>
> | Factorization | DS | GPTScore | MMD |
> |-|-|-|-|
> | (1) | 0.32 | 0.67 | 0.48 |
> | (2) | 0.32 | 0.68 | 0.50 |
> | Ours* | 0.34 | 0.70 | 0.48 |
>
> ## 2. Choice of time distribution
> We confirm that modeling $\Delta t_i$ with an exponential-family distribution is a design choice, not a restrictive assumption. This choice is standard in TPP literature (e.g., RMTPP), as it enables tractable likelihood computation and efficient sampling.
>
> Importantly, FC-TPP is agnostic to the specific time distribution. Any positive-support distribution (e.g., power-law, or more flexible neural parameterizations) can be used by modifying the decoder, without affecting the reasoning module.
>
> Empirically, we find that performance is not sensitive to the specific distribution choice, and is primarily driven by the latent reasoning mechanism. We therefore adopt the exponential family for simplicity and stability.
>
> ## 3. Choice of reasoning depth $H$
> In the current design, $H$ is a small fixed hyperparameter (e.g., $H\leq$ 10), chosen based on validation performance. Empirically, we observe that performance saturates quickly with increasing $H$, indicating that shallow reasoning is sufficient in practice.
>
> Follow suggestion by **Reviewer x12A**, we also explored a dynamic stopping criterion, where reasoning continues until convergence (i.e., $||v^{(H+1)}-v^{(H)}|| \leq \epsilon$). On MIMIC-IV, this yields slight improvements (MMD: 0.48 → 0.45) by adaptively adjusting reasoning depth, but at the cost of increased computation.
>
> In practice, we find that the fixed-hop design provides a trade-off. It achieves comparable performance while being simpler, more efficient, and more stable (e.g., avoiding vanishing gradients). Therefore, $H$ can be selected as a small constant, with dynamic variants as an available optional extension.
>
> | $H$ Schedule | Avg. $H$ | DS | GPTScore | MMD |
> |-|-|-|-|-|
> | Dynamic | 5.82 | 0.32 | 0.70 | 0.45|
> | Fixed | 5 | 0.34 | 0.70 | 0.48 |
>
> ## 4. Use of raw event history in the decoder
> The decoder is not independent of history. The event history is first encoded into the recurrent state $z_t$, and then mapped and refined into $v_t^{(H)}$. Thus, the decoder conditions on a history-informed latent state, rather than raw events.
>
> Conditioning only on $v_t^{(H)}$ is intentional. It enforces a decoupling between raw history encoding and generation, so that event generation is driven by a reasoning-completed symbolic state rather than directly by noisy or long histories. This design improves controllability, as modifying predicates or rules leads to predictable changes in generation; directly conditioning on $z_t$ would re-entangle generation with raw history and weaken this control. Moreover, $v_t^{(H)}$ acts as a compact, structure-aware summary of history, filtering noise and capturing high-level dependencies, which is beneficial for long or partially observed sequences.
>
> That said, the framework is flexible: the decoder can be extended to condition on both $v_t^{(H)}$ and $z_t$ if finer-grained temporal information is needed. This modification is straightforward and does not affect the reasoning module.

---

> > ### Author Rebuttal · Reviewer_pMnX · 2026-04-04
> >
> > Thanks authors for their response. It might be beneficial to revise the formulation in the paper to explicitly include depedency between event type and event time.
> >
> > I will keep my score as 4.

---

> > > ### Author Response · Authors · 2026-04-05
> > >
> > > We sincerely thank the reviewer for the helpful suggestion and positive assessment.
> > >
> > > (i) We will revise the formulation to explicitly include depedency between event type and event time.
> > >
> > > (ii) Inspired by your quesiton on whether to incorporate history embeddings, we design additional experiments to directly address this beyond the conceptual discussion.
> > >
> > > **Setup**. We generated synthetic event process by introducing some **rare events** driven by latent rule state $R_t = 0\ \text{or}\ 1$, defined via multievent combinatorial condition over a long time window (*not recoverable from short-term history*). Such as $R_t \leftarrow A \wedge B \wedge C$, where $A, B, C$ are separated by long gaps. Background events follow a multivariate Hawkes process (strong local correlations). Target intensity:
> > >
> > > $$
> > > \lambda_Y(t)=\mu_Y+\eta_R  \mathbf{1} (R_t=1)+\eta_H H_t
> > > $$
> > >
> > > The synthetic event sequences contain 10 event types in total, including 2 rare event types. We use 5000 sequences, with an average of ≈33 events per sequence. We consider 1) **FC-TPP** (ours, symbolic $z_{\text {sym}}$ as state), 2) **THP** (history $h_{\text {hist }}$ as state), 3) **Naive hybrid** ($\operatorname{concat}\left(z_{\text {sym }}, h_{\text {hist }}\right)$ as state), 4) **Gated hybrid**, where **FC-TPP and THP produce separate predictions and the final output is selected via a simple confidence-based gating mechanism**.
> > >
> > > The next-event prediction results (across all event types, with rare events accounting for approximately 13% of true events) are:
> > >
> > > | Model | MAE | ER$\%$ |
> > > |-|-|-|
> > > | FC-TPP | 2.15 | 17 |
> > > | THP | 2.57 | 25 |
> > > | Naive Hybrid | 2.22 | 17 |
> > > | Gated Hybrid | 2.08 | 16 |
> > >
> > > The results show that (1) the **Naive hybrid** does not consistently improve performance in the above designed rare-event settings, and (2) the **Gated hybrid** achieves the best (slightly better than FC-TPP). This highlights that history information should be integrated **in a controlled way** rather than through direct fusion. Our FC-TPP retains a key advantage: it **explicitly isolates a symbolic rule-based core for interpretability**, while **remaining readily extendable to hybrid designs**. We will further elaborate on these extensions in the revised paper.

---

### Decision · Program_Chairs · 2026-04-30

**Decision:**

Accept (regular)

**Comment:**

FC-TPP introduces a reasoning-before-generation framework for temporal point processes, where a predicate-level latent state evolves via differentiable multi-hop forward chaining and drives event generation. Reviewers praised the clean architectural principle, the integration of symbolic reasoning with continuous-time generation, and the comprehensive empirical evaluation across synthetic and real-world benchmarks. The rebuttal meaningfully addressed concerns around time-type dependency, sensitivity to rule quality, reasoning-depth choice, and standard TPP metrics, including new experiments showing recovery from noisy rule initialization. I recommend acceptance.